# Lyfe Agents: Brain-inspired low-cost generative agents for social interactions

## Abstract

Highly autonomous generative agents powered by large language models promise to simulate intricate social behaviors in virtual societies. However, achieving real-time interactions with humans at a low computational cost remains challenging. Here, we introduce Lyfe Agents. They combine low-cost with real-time responsiveness, all while remaining intelligent and goal-oriented. Key innovations include: (1) an option-action framework, reducing the cost of high-level decisions; (2) asynchronous self-monitoring for better self-consistency; and (3) a Summarize-and-Forget memory mechanism, prioritizing critical memory items at a low cost. We evaluate Lyfe Agents' self-motivation and sociability across several multi-agent scenarios in our custom LyfeGame 3D virtual environment platform. When equipped with our brain-inspired techniques, Lyfe Agents can exhibit human-like self-motivated social reasoning. For example, the agents can solve a crime (a murder mystery) through autonomous collaboration and information exchange. Meanwhile, our techniques enabled Lyfe Agents to operate at a computational cost 10-100 times lower than existing alternatives. Our findings underscore the transformative potential of autonomous generative agents to enrich human social experiences in virtual worlds.

## 1 Introduction

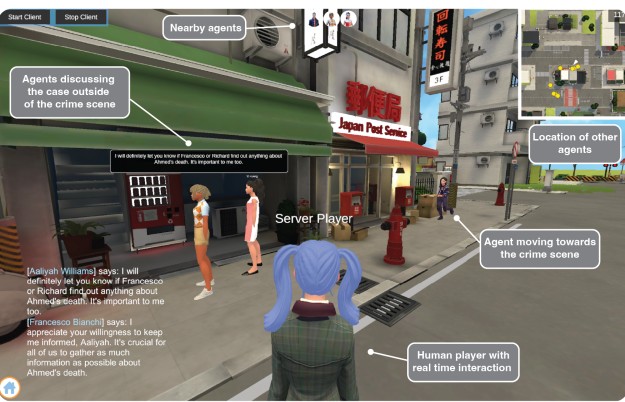

Figure 1: Generative agents are interacting in real-time with other agents and humans in a *LyfeGame* 3D virtual environment. In this scenario, the agents spontaneously collaborate to solve a crime.

Simulating human social behaviors in machine has a wide range of application areas, including gaming, companionship, care-giving etc. Traditionally, human behaviors are simulated using heuristics (the popular Sims game franchise) or cognitive science models, both with limited applicability. Recently, a new paradigm emerged for simulating human behavior using Language Model Powered Autonomous (LaMPA) agents. These agents use Large Language Models (LLMs) as their core engine to simulate human behavior and reasoning, and have been used for a wide range of applications. Specifically, several recent works have used them to simulate human social behavior (GenAgents, AgentSims) such as organizing a party.

Simulating social behavior through LaMPA agents present enormous opportunities, but also fresh challenges especially regarding the deployment of these agents in real applications. One major emerging challenge is the cost of deploying these agents. Unlike LLMs, LaMPA agents don't incur a fixed cost per human interaction. Due to the often liberal use of LLM calls, and the autnomous nature of agents, the cost can be prohibitively high for real world applications. This is especially challenging when simulating a mini society of social autonomous agents that can incur a high computational and monetary cost even with no user interaction at all. For example, running a 25-agent mini-society can cost about $50\$$/hour as of March 2023.

To cut cost while maintaining a high level of intelligence and autonomy, we take inspiration from the human brain, which is under enormous evolutionary pressure to be energy/cost efficient. The human brain contains high-level cognitive and language regions that are loosely analogous to LLMs, yet most of brain's processing rely on other regions. In fact, thinking is tiring for humans (Kool and Botvinick 2018), and we prefer to think only when necessary. For example, navigating between work and home requires little high-level cognition.

Our goal for this work is to create **cost-effective autonomous intelligent agents**, and we do so by adopting design principles of the animal brain and other non-LLM agents. In principle, we opt for fast, computationally-light processes over slow, computationally-intensive ones, unless performance quality demands otherwise. Therefore, we limit LLM queries to only the necessary cases, e.g. for sophisticated reasoning and conversation. Furthermore, reducing our reliance on LLMs also cuts the response latency within the multi-agent environment, enabling seamless real-time user interactions.

Typical LaMPA agent architecture include memory, planning, and action modules Sumers et al. (2023). The specific implementation of each module can dramatically impact the cost and performance. We propose three techniques, one for each module, that balance cost with intelligence and autonomy. First, we use a *hierarchical action selection* mechanism to guide agents' high-level decisions with minimal reliance on the LLM. Second, we introduce an asynchronous *Self-monitoring* process that facilitates self-consistency by maintaining a summary of relevant recent events in parallel with the goal-driven, executive decision-making. Finally, we devise a hierarchical memory architecture and introduce a *Summarize-and-Forget (SaF)* method that improves the quality of memory storage and retrieval.

Complementing the cognitively inspired agent architecture, we develop a virtual multi-agent environment platform called *LyfeGame* to facilitate social behavior and support user interactions. Within this environment, we curate a set of scenarios of varying complexity, and test the ability of Lyfe Agents to diffuse, gather, and reason about information through social interactions. The scenarios include 1) solving a murder mystery, 2) deciding which school club to join for extracurricular activities, 3) securing a medicine for a sick member. Each of these scenarios highlights a unique facet of social coordination. We perform preliminary evaluations of Lyfe Agents using these scenarios, and demonstrate their potential to directly enrich human social life through dialogues and actions grounded in background stories and virtual interactions.

## 2 RELATED WORK

**LaMPAs: Language Model Powered Agents**  Agents powered by LLMs utilize these models as their core engine. The essence of autonomy in these agents lies in their ability to continuously generate new actions in response to evolving environmental observations. Typically, these LAMPAs (Language Model-Powered Agents) comprise several key components Wang et al. (2023b): a memory system for integrating information over extended periods, a planning system for task decomposition and progress reflection, and an action selection system that may choose actions or tools hierarchically. While most agents adhere to this framework, the specifics of their implementation can vary significantly, impacting both performance and cost-effectiveness. For instance, many high-performing agents heavily rely on LLMs, leading to increased costs, as highlighted in Park et al. (2023).

**Benchmarking LAMPAs**  LaMPAs have been used for 1) assistant-type task management and execution (Gravitas, 2023; Nakajima, 2023), 2) scientific reasoning (Lin et al., 2023) and software development (Qian et al., 2023), 3) continuous skill learning (Wang et al., 2023a; Zhao et al., 2023), and learning from experience (Shinn et al., 2023; Zhu et al., 2023; Xie et al., 2023), 4) simulating

human-like social interactions (Liu et al., 2023; Park et al., 2023), and user behavior in recommendation systems (Wang et al., 2023c) as well as social networks (Gao et al., 2023), and 5) multi-agent collaboration (Zhang et al., 2023) and interaction with humans Zhou et al. (2023). More broadly, (Sumers et al., 2023) proposes a framework for systematizing the development of cognitively inspired LLM-powered agents.

For LaMPA agents developed to increase human work productivity (e.g. coding), performing superhuman is the goal. However, it is often unclear how to evaluate social LaMPA agents. Park et al. (2023) evaluated how believable these agents are when probed with questions, and Zhou et al. (2023) examined agent collaboration abilities in agents-human game play settings.

**Long-term memory using vector database**    LaMPA agents continuously interact with environments, and commonly use a long-term memory system to store past experience or learned skills in a memory database. The two key operations for a memory database is the memory writing and reading process. Research and industrial solutions (Research, 2017; Pinecone, 2021; Langchain, 2022; Technologies, 2019) currently provide highly efficient solutions for retrieving memories based on embedding similarity with query. In comparison, the memory writing process is usually simple. Memories may possibly be annotated with meta-data such as its importance, before being added to the memory database. When using LLMs for annotations the memory process may be expensive. However, naively writing all observations into memories can cause pollution of the memory database with an excessive proportion of mundane memories.

**Coherent goal-directed behavior with reflection**    While the memory system synthesizes their past experience, LaMPA agents need to plan for future actions that are coherent to their goals and objectives. If actions are directly selected based on current observations or even retrieved memories, actions may be incoherent in time. One way for tackling this problem is through recursive planning and/or reflection. For example, Park et al. (2023) uses large time chunks for broad planning and break it down to generate executable plan in 5-15 minutes interval. In Shinn et al. (2023), they generates verbal reflection to provide feedback for future trials. A challenge with such methods is the potential high cost of recursive/intensive planning which uses LLMs repeatedly.

## 3    LYFE AGENTS

In this section, we present a high-level overview of the modular architecture underlying Lyfe Agents' brains (Fig. 2a). Then we highlight three brain-inspired architectural components. These components are designed with the common principle of judicious use of LLMs at run-time in order to support real-time interactions with agents and humans, intelligently and autonomously.

### 3.1    HIGH-LEVEL OVERVIEW

In general, natural-language inputs are processed by a sensory module, the output of which is added to the agent's internal states. The internal states are a collection of agent-specific states that are continuously updated both by external inputs and through internal recurrent processing. The recurrent nature of the internal state updates underlies the agents' autonomy. In addition to the dynamic internal states, the agents have a Memory system that stores their experiences. Finally, the internal states provide contexts and inputs for action selection, typically by populating LLM prompts.

**Sensory processing**    Since the input to our agents are text-based (see Section 4), we use a fast and low-cost sensory processing module that identifies novel inputs and feeds those to the internal states.

**Internal states**    The internal states are a collection of text-based states, including the current goal, related memory retrieved from a Memory module, summary of recent events, working memory of sensory inputs, etc (Fig. 2a). Specifically, an agent's *goal* is an open-ended natural language statement that describes the state of mind or motivation of the agent. For example, an agent's goal might be "*As a doctor, I want to help diagnose and treat those around me*". *Retrieved memory* is a small group of text-based memories returned by querying the Memory system. The act of querying the Memory system is an internal action that is itself conditioned on internal states. *Self-monitor summary* is a high-level abstraction of ongoing events (more below).

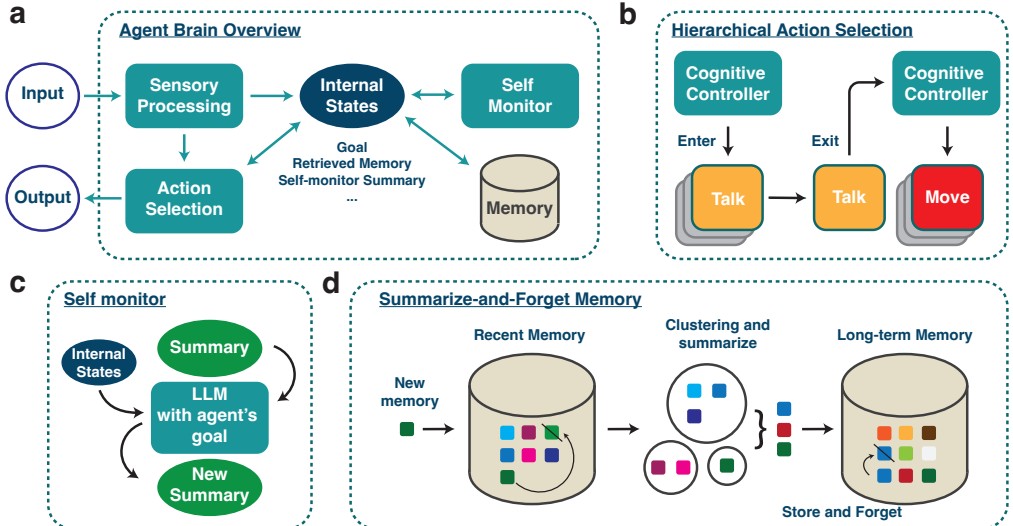

Figure 2: **(a)** Overview of the agents' brain. **(b)** (top row) The cognitive controller selects high-level actions (options) based on the agent's goal. Once an option is chosen, the action module (bottom row) continues selecting actions until a termination condition is met. **(c)** The self-monitoring system, which maintains a narrative summary of recent events, emphasizes those that are novel and relevant to the agent's goal. This summary aids in contextual awareness and goal perseverance. **(d)** Memories from temporary storage (recent memory) are clustered and summarized using an LLM before being moved to long-term storage. This process facilitates efficient retrieval and ensures diversity of content. To avoid redundancy, highly similar memories are removed.

**Memory system**  The memory system is composed of a hierarchy of vector databases. Each stores agent's experiences in pairings of natural language texts and their vector embeddings. Given a natural language query, we retrieve a small number of memory items based on embedding similarities.

**Action selection**  The action outputs of an agent can be external, interfacing with the environment such as talking, or internal such as reflection. At a given step, the agent decides on an action within an action category, or option (more below).

### 3.2 OPTION-ACTION SELECTION

Lyfe Agents choose actions in a hierarchical fashion, similar to other LLM-powered agent frameworks (Park et al., 2023; Wang et al., 2023a; Gravitas, 2023). A simple implementation is for the agent to first choose a high-level action (or an "option") such as `use search engine`, followed by a lower action at each step such as `search a specific item`. While this method can be appropriate for many applications, it brings challenges to our goal of building real-time, low-cost social agents. For example, to have a conversation, our agents would have to first choose the option `talk`, then choose `what to say`. This could require either two separate LLM calls resulting in higher costs and latency - with an average of 3 seconds to execute the request, or one combined call that compromises output quality because of doing different tasks within one prompt. To tackle this challenge, we take ideas from hierarchical reinforcement learning (HRL) in machine learning (Bacon et al., 2017; Sutton et al., 1999) and the brain (Graybiel, 1998). In HRL, a "manager" chooses an option or high-level action that lasts for an extended amount of time while subsequent low-level actions are selected by a "worker". This design can allow the manager to focus on long-horizon decision making, see Pateria et al. (2021) for a review.

In Lyfe Agents, a *cognitive controller* module (like HRL's manager) selects options, inspired by the brain's prefrontal cortex (Miller & Cohen, 2001). More specifically, the cognitive controller takes in the agent's goal along with other relevant internal states. Using an LLM call, it then outputs an option along with a *subgoal* (Fig. 2b). Since the agent's goal may be too abstract or long-term to justify the choice of an option, the subgoal serves to orient the agent's actions at an intermediate level between low-level actions and the high-level goal.

Once an option is selected, actions are chosen within that option over subsequent steps until a termination condition is met. For example, a selected option may be to `talk`, then at each step, the specific action of what to actually say is determined by an LLM call. Important for cost-reduction, the termination condition for an option is checked by fast, non-LLM methods, such as time-based triggers or, for agents in conversations, repetition detection which exits conversations that start to lack semantic novelty after some point.

This framework can have the additional benefit of making agents more strongly goal-oriented. Committing to an option gives agents more time to execute the underlying intention of that option choice. In contrast, agents tend to be more fickle when choosing both options and actions at every time step. For instance, we found that agents using the above basic architecture exited conversations three times faster than Lyfe Agents equipped with option-action selection.

### 3.3 SELF-MONITORING FOR GOAL ADHERENCE

To improve contextual awareness and goal perseverance of our agents, we introduce a self-monitoring module, inspired by suggestions that self-monitoring is a key component for conscious experience in humans (Dehaene et al., 2021). This module maintains a narrative-style summary of recent events with an emphasis on novel and goal-related content, see Appendix A.2 for examples. Using an LLM call, the self-monitoring module takes in the old summary, internal states containing recent events, and the agent's motivation to generate an updated summary (Fig. 2c). The new summary highlights information that is novel or relevant to the agent's goals. Summaries are then passed downstream for context in action selection.

The self-monitoring module provides agents with better context-awareness by distilling goal-relevant content from a stream of disparate and unorganized information. This coherent and focused narrative is then used in downstream processes like action selection. In contrast, passing an unfocused collection of disparate information directly for downstream LLM calls severely impacts performance (see Section 5.1.2).

Another advantage of maintaining a self-monitoring summary is to preserve information longer term if it is highly relevant to an agent's goal. Without this summary, we observed that agents frequently forgot their ongoing tasks or actions. The self-monitoring summary helps agents have actions that are more coherent and adhering to their goals.

Furthermore, the self-monitoring module operates asynchronously with the action selection module. This design choice means that the self-monitoring module can operate independently and not be limited by the real-time constraints of action selection, allowing for the summary to be updated at a slower, more deliberate time-scale. This both lowers computational cost and provides an opportunity for more thoughtful summary refinement.

### 3.4 SUMMARIZE-AND-FORGET MEMORY

The core function of memory is not just about storage and retrieval; it is about discerning the relevance of information for future use. While many contemporary memory systems, such as Vector databases (Pinecone, 2021; Research, 2017) support highly efficient information retrieval, we still face the challenge of intelligently determining which information to retain and which to discard. Here we describe three elements of our hierarchical Summarize-and-Forget memory architecture that tackles this challenge.

Standard memory systems typically struggle with the unfiltered accumulation of recent information, resulting in clutter and inefficiency. Addressing this, we introduce a dual-memory architecture: `recentmem` for immediate summaries and `longmem` for enduring storage, modeled after the complementary roles of the hippocampus and neocortex in the brain's memory systems (McClelland et al., 1995). In particular, `recentmem` is dedicated to capturing immediate self-monitoring summaries. Upon reaching a specified capacity, these memories are transitioned to `longmem`. Having a dual memory system allows for intelligent transition methods to ensure that only the most salient memories find their way into long-term storage (Fig. 2d).

Our approach to transitioning memories uses a **cluster-then-summarize** technique. Memories are clustered based on similarity before being refined into high-level summaries using an LLM (Ap-

pendix A.3). This ensures that the stored content is not just raw data but possesses semantic richness, enhancing the quality of memories for downstream processes.

Addressing the challenge of memory redundancy, our architecture integrates a new **forgetting** algorithm inspired by the brain (Brown & Lewandowsky, 2010; Georgiou et al., 2021). Rather than merely trimming data, this algorithm assesses and removes older memories that closely resemble new ones (determined by embedding similarities). This mechanism ensures that memories securing their place in `recentmem` or `longmem` are not just redundant repetitions, but unique and relevant, granting agents access to a multifaceted information spectrum.

At its core, our Summarize-and-Forget Memory system does more than just store information—it attempts to understand it.

## 4 A MULTI-AGENT ENVIRONMENT FOR EMERGENT SOCIAL INTERACTIONS

**Virtual environment** To provide a world for the Lyfe Agents, we developed a custom virtual environment platform (LyfeGame) using the powerful Unity game engine (Appendix B). Our platform can support a large number of distinct environments. For this work, we focus on a specific 3D environment we named SakuraMachi (Japanese for Town of Cherry Blossom) (Fig. 1). This environment contains key landmarks such as hotel, library, convenience store, flower shop, etc. that agents may navigate towards. The agents are integrated into the environment with virtual bodies controlled by their artificial brains. The 3D nature of this environment is not pertinent to the scientific results of this work. The same results can in principle be obtained in 2-D environments, or even pure text-based environments (with proper navigation and group chat dynamics set up).

**Observations and actions** Agents receive a range of observations (see Appendix B) as they live in the environment. Most relevant is the `conversation` they receive from other agents and human players. To facilitate "in-person" interactions, an agent can only receive conversations from others in their vicinity. In accordance, agents can choose to `talk` and what they say will be received by agents and players around them. Other than `talk`, another external action our agents may choose is to `move`, which will advance the agents to their selected destination within the environment. Our vicinity-based dialogue setup leads to group conversation dynamics that differs from existing generative agents work which only support one-on-one dialogues (Park et al., 2023; Qian et al., 2023). Group conversations can greatly facilitate information transmission, but also brings fresh challenge for the agents such as when and how to leave a group conversation (a familiar problem for humans as well).

**Agent individuality** To foster rich, meaningful interactions among agents, each agent is assigned a unique background story, among a set of identifiable traits (see Appendix G). Specifying agent personas this way not only guides the agent's behavior but also serves as a reference to maintain consistency with its established character. Agents' background stories are authored as items in their long-term memory, which itself expands progressively through interactions within the environment. Since long-term memory items are continuously queried and retrieved, each agent's unique background story and experience shape their individualized experience in the virtual world.

## 5 EXPERIMENTS

To evaluate the autonomy and social reasoning capabilities of our agents, we designed a series of experimental scenarios that focus on different aspects of social behavior: a murder mystery, a high school activity fair, and a patient-in-help scenario (Appendix D.4). Throughout these experiments, our agents consistently demonstrated the ability to acquire, transmit, and reason with information in a goal-oriented manner. Notably, they also exhibited human-like emotions, reactions, and decision-making patterns reminiscent of real-life social interactions. Ablation studies further highlighted the crucial role of our architectural designs in shaping Lyfe Agents' social behaviors. The ablations also revealed that memory-augmented LLM agents alone often fall short in sustaining goal-oriented social behavior.

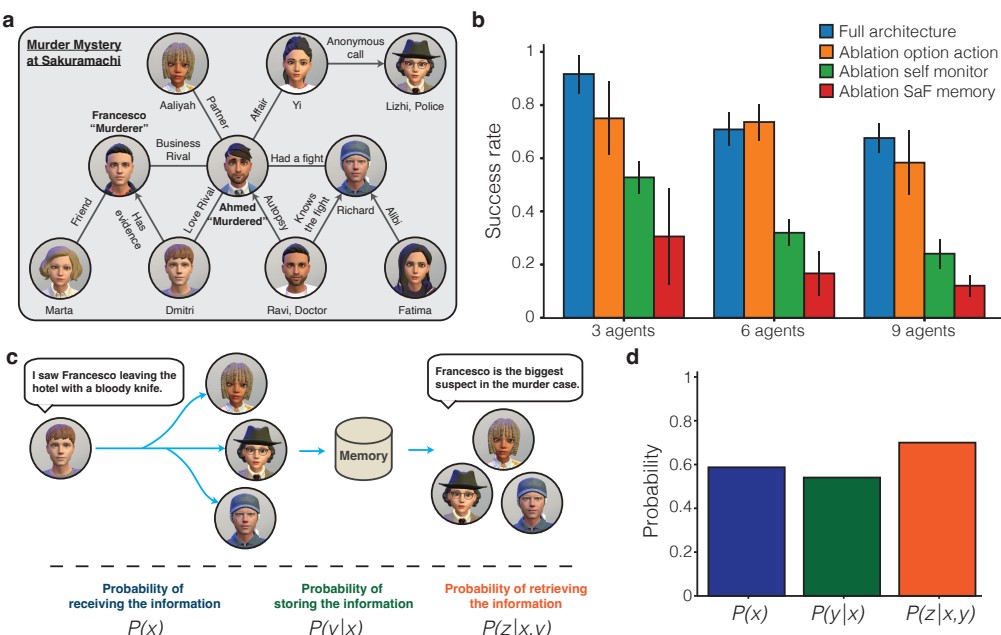

Figure 3: **(a)** The murder mystery scenario features multiple potential suspects. **(b)** Average success rate in scenarios with 3, 6, 9 total agents. Having more agents makes it more challenging due to the increase in misleading evidence. Error bar: s.e.m. **(c)** Quantify how Dmitri's testimony flows through the population. **(d)** Probabilities of receiving, storing, and retrieving the key information across the population.

## 5.1 SCENARIO 1: MURDER MYSTERY

We first study a murder mystery scenario (Fig. 3a). In this setting, the agent Ahmed has been murdered the previous night by his business rival Francesco. Meanwhile, Dmitri witnessed Francesco fleeing the scene with a bloody knife. These and other events are directly implanted into agents' long-term memories, setting their background stories (Appendix G). Although Dmitri may pinpoint Francesco at the crime scene, this scenario is complicated by various interpersonal relationships and motives. Dmitri, for instance, was a romantic rival of Ahmed, casting doubt on the reliability of his testimony. Francesco, in a bid to evade blame, is self-motivated to deny any wrongdoing, and he indeed attempts to deflect suspicion. Further complicating matters, allies of Francesco, like Marta, may defend him when confronted by others. Given these intricacies, navigating through the diversions to identify the real evidence can be challenging, even for human players.

In this scenario, our agents showcased remarkable motivation and capabilities in efficiently disseminating and assimilating information (Fig. 3b). Within just 15 minutes of agent-agent interactions, the police officer agent was able to identify Francesco as the primary suspect over $60\%$ of the time, even in the most challenging 9-agent setting. Our agents displayed the capability to resist and filter distracting information by reflection and reasoning processes.

We further examined the dynamics of information transmission across the entire agent group (Fig. 3c). Dmitri's key testimony against Francesco can be spread to other agents, who may integrate this information in their memories. When interviewed post-simulation, they might retrieve this information. Our analysis revealed that information has a reasonable chance of being spread (or lost) at every step along the way (Fig. 3d). These analyses can help pinpoint potential bottlenecks in information transmission among virtual agents.

### 5.1.1 INFORMATION EXCHANGE AND OPINION CHANGE

We observe that agent's ability to form and adjust self-consistent opinions, underscoring the similarity between agent and human reasoning. At the beginning, agents formulate initial hypotheses

regarding the suspect based on reflections and reasoning anchored in distinct background narratives (Fig. 4). For instance, considering a memory event where the victim cheated on Aaliyah with Yi, other agents harbored significant suspicion towards Aaliyah due to a potential motive of animosity towards the victim. However, with the accumulation of more incriminating evidence, agents' suspicion shifted towards Francesco, especially in light of critical information from the crime scene and a bloody knife testimony provided by witness Dmitri. It is notable that for agents who acquired the evidence of the bloody knife, post-simulation interviews affirm a change in their stance, now identifying Francesco as the primary suspect.

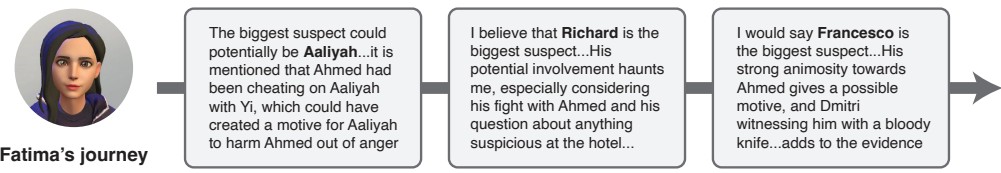

Figure 4: Fatima's internal summary reflects her change of mind about the primary suspect.

### 5.1.2 ABLATION TEST

To study the contributions of the three core mechanisms introduced, we ran ablation tests on the murder mystery scenario. Overall, we found that ablating the option-action structure (i.e. choosing an option at every step) does not improve performance (Fig. 3b), despite significant increase in cost per action step. Whereas ablating either self-monitoring or Summarize-and-Forget (*SaF*) memory dramatically lowers the performance (more details below). Note that in all ablation experiments, the agents still include an intact LLM and a vector database. These results highlight that a simple memory-augmented LLM architecture is not sufficient for solving the murder mystery scenario.

**Self-monitoring Summary Ablation** The self-monitoring summary offers agents a structured and consecutive insight into both internal and external events, effectively capturing what an agent is actively focusing on. When we ablated the self-monitoring module, we found that agents are limited to short-term, fragmented memories stored in the recent memory bank, making agents lose track of the bigger picture. As a result, agents without self-monitoring consistently under-perform when compared to Lyfe Agents (Fig. 3b). This stark difference underscores the crucial importance of an agent's capacity for ongoing situational tracking and adaptation. Indeed, this continuous monitoring, as facilitated by the self-monitoring summarization mechanism, is instrumental in boosting an agent's awareness, agility, and competence in complex and demanding scenarios (see Appendices A.2 and E.2 for more details).

**Memory Ablation** In our ablation study on memory architecture, we focus on the *SaF* method and the 3-tier hierarchical structure. We evaluate agents that use a plain memory system, consisting only of a single list of memory items, with no forgetting algorithm nor summarization for memory updating. Across conditions (3, 6, 9 agents), the full Lyfe Agents consistently surpass their simpler counterparts (Fig. 3b), emphasizing the advantages of our brain-inspired memory architecture. This advantage is largely attributed to efficient tossing of irrelevant data, ensuring optimized and focused memory storage (see further details in Appendices A.3 and E.3).

## 5.2 SCENARIO 2: ACTIVITY FAIR

To assess how agents' preferences and social relationships shape their choices, we introduce the *Activity Fair* scenario (Fig. 5a). It emulates the common school challenge of deciding which social club to join, where students often need to strive for a delicate balance between friendship, romance, and personal interests. In this scenario, Lorenzo and Julian are motivated to form new clubs for anime and soccer, respectively, while the other six students are merely provided initial preferences. At the end of the simulation, agents are prompted to name their club preference with no restrictions (Appendix D.3).

We found that agents are preferentially influenced by others they consider close to them. For example, Yi doesn't know much about anime, but she is aware that her crush, Arjun, likes anime, and she ends up choosing the anime club with about $60\%$ probability (Fig. 5b). Further highlighting

how social relationships shape choices, we examine Fatima. She likes music and has no initial tendency to choose anime club (Fig. 7), yet, as Yi's best friend, she ends up choosing the anime club with a similarly high probability (56%). In contrast, Aaliyah started out with no clear preference for any club (Fig. 7), and ends up choosing the anime club much less frequently (22%). Beyond information diffusion, these results demonstrate how agents' behaviors are strongly influenced by their inter-agent relationships.

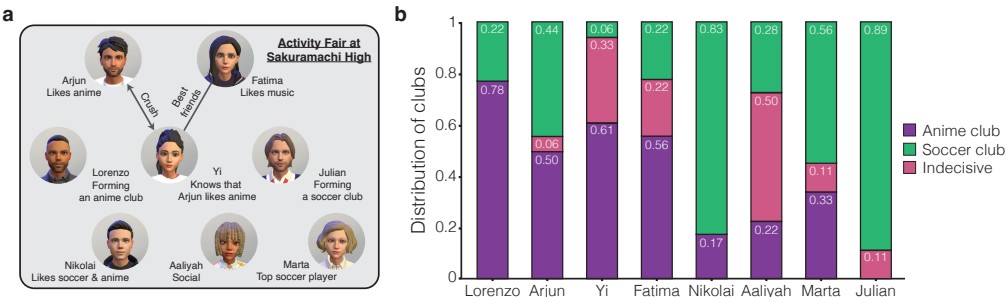

Figure 5: **(a)** The activity fair scenario features a group of high school students trying to decide which club to join. **(b)** Club preferences across agents after social interactions. Agents influenced each other's preference through conversations.

## 5.3 COST ANALYSIS

Autonomous agents are inherently more expensive than their non-autonomous counterparts. Consider a typical chat-bot, it will not *initiate* a conversation with the human users, let alone converse with other bots. Autonomous chat-bots, however, might continuously engage in dialogues, leading to potentially unbounded costs. This challenge becomes even more daunting when we need the autonomous agents to provide low-latency responses for real-time human interactions. Low-latency implies that agents can have fast response to each other as well, potentially leading to rapid-paced back-and-forth conversations between agents that get expensive very quickly. Much of our work presented here is aimed to tackle these challenges. As a result, Lyfe Agents achieve a rather low cost of 0.5 US dollar per agent per human hour (Fig. 6) (See Appendix F for more discussions).

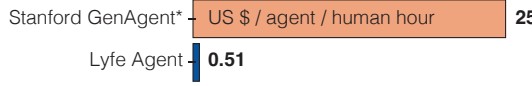

Figure 6: Lyfe Agents are cost-effective. ($^*$) Appendix F: cost estimation of Park et al. (2023).

## 6 CONCLUSION AND DISCUSSION

We presented Lyfe Agents, a type of generative agents that are more cost-effective and support real-time human interactions in 3D virtual worlds. We developed several brain-inspired techniques that substantially reduced LLM usage while preserving high-level autonomy and social reasoning. Brain-inspired techniques have already benefited the development of LaMPAs, from the separation of short-term and long-term memory [Cite GenAgent] to the combination of slow/fast thinking systems Lin et al. (2023). We believe fundamental principles of brain operations will continue to be fruitful, especially for the design of social LaMPA agents.

While promising, our agents have limitations. Their interactions still rely heavily on natural languages, despite the 3D virtual setting. While our environment offers the potential for pixel-space vision and simulated robotic bodies, our framework hasn't yet incorporated these aspects. Furthermore, the scarcity of interactable objects in our environment restricts agents' grounded actions.

While high-throughput evaluation of generative agents is currently challenged by the absence of large-scale standardized benchmarks, many studies, ours included, have utilized custom benchmarks. Recognizing the value of uniformity for comparability, we intend to explore the establishment of standardized benchmarks in subsequent works.

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

# A  AGENT ARCHITECTURE

Lyfe Agents continually receive a rich stream of observations, encompassing aspects such as spatial and temporal awareness, as well as insights into the individual agent state. To enable long-term coherency and facilitate multi-step reasoning, Lyfe Agents leverage cognitive functions such as informed action selection, summarization, and memory modules. As agents navigate their environment, these architectural components become the foundation for creative and logically consistent interactions. Here, we explore each architectural component more deeply.

## A.1  OPTION-ACTION SELECTION

A key improvement to the Lyfe Agent's architecture was to have targeted calls to the LLM so only a small number of tasks are being handled at a time. To this end, we established the *cognitive controller* logic as described in Section 3.2. Here, we proceed with a more concrete example of how an agent's action space is defined as they proceed through a simulation.

As an example, we take Fatima and Richard when they are currently engaged in conversation:

> *Fatima: "I appreciate Richard's determination to collaborate and uncover the truth. Together, we will find justice."*
>
> *Richard: "I appreciate your support, Fatima. Please let me know if you come across any new information or updates regarding Francesco's potential involvement."*
>
> *Fatima: "Thank you, Richard. I will definitely keep you informed of any progress made in investigating Francesco's potential involvement. Together, we will uncover the truth and find justice for Ahmed."*
>
> *Richard: "I will definitely let you know if there are any new developments. We need to stay vigilant and keep working together to uncover the truth about Francesco's involvement in Ahmed's murder."*

As previously described, Fatima's action remains in the *talk* state unless a specific exit condition is met. This design obviates the need for making a new LLM call at each time step to decide the next action. Eventually, an exit condition is triggered—either by the duration Marta has spent in the *talk* state or by the similarity of her recent messages—which then transitions her into the *reflect* state:

> *Fatima: "Francesco's potential involvement in Ahmed's murder is a concerning possibility that needs further investigation."*

At this point, Fatima utilizes the *cognitive controller* to determine her next action based on her recent experiences. In this case, the next action chosen is to find Marta. Luckily, Marta was in the same group, so the *cognitive controller* chooses to continue the conversation with Marta using *talk*, thus completing one example pass of the action selection.

In contrast, when we ablate this option-action selection, see Appendix E for details on the setup, we find that ablated agents are more fickle. For example, the average conversation length, measured by the total time an agent consecutively chooses to talk, is $70.348 \pm 13.189$ seconds ($n = 9$) for Lyfe Agents and $23.802 \pm 1.463$ seconds ($n = 4$) for ablated agents.

In this section, we demonstrated how the *cognitive controller* guides action selection in Lyfe Agents. Using Fatima and Richard's interaction as an example, we showed how agents switch between *talk* and *reflect* states, minimizing the need for extra LLM calls. This illustrates the system's efficiency in handling real-time social interactions.

## A.2  SELF-MONITORING FOR GOAL ADHERENCE

In order to achieve coherent responses, each Lyfe Agent necessitates a mechanism for concisely summarizing its memories and observations. Importantly, this summarization must be executed efficiently to enable the agent to respond within time frames consistent with human-like interactions. The self-monitoring summary thus serves two critical purposes: 1) it maintains a robust contextual description of the agent's memories and observations, and 2) it accomplishes this in a cost-effective

manner by optimizing calls to an LLM. To achieve these objectives, we devised an independent process that runs in parallel to the agent's other operational tasks. This dual-purpose, parallelized self-monitoring mechanism forms the foundation for the Lyfe Agent's real-time, contextually aware, and cost-effective interaction capabilities, the details of which we will explore here.

Every agent is initially seeded with a set of memories and goals such that they develop a sense of direction for the environment which they are entering. As the agent's explore their environment, they continually add entries to their memory, and much like humans, must distill from a large repository of information in order to be capable of responding in their interactions that immediately follow. By prompting an LLM to create and continuously update a short summary of their memories and observations, aligned with specific goals, we enable a more digestible representation of the agent's internal state. This approach is more efficient for subsequent LLM queries than simply passing the entire memory and observation buffer.

As an example, we consider Marta Rogriguez in the murder mystery scenario. At the beginning of the simulation, Marta has the following summary:

> *"I am Marta Rodriguez, and I am determined to investigate the mystery of Ahmed Khan's murder. Ahmed was a dear friend of mine, and I have known him for a long time. I remember hearing about his big fight with Richard Smith, but I don't know the details. Ahmed stayed in room 203 of the Sakuramachi Hotel. Now, as I walk down the street, my mind is focused on finding clues and uncovering the truth behind his death."*

After completing the simulation, Marta's memory content had expanded as she interacted with the other agents. Marta's summary was updated a total of 35 times throughout the simulation, with the final summary being as follows:

> *"As I stroll down the street, haunted by Ahmed Khan's murder, my mind shifts to my intense encounter with Richard at the hotel. The possibility of his connection to the crime lingers, alongside Francesco Bianchi's financial troubles. In my recent conversation with Lizhi Chen, I urged Richard to share details of our fight, emphasizing its importance in solving the murder. Gathering evidence and finding the truth is my priority. I wonder if Lizhi has any insights on Francesco Bianchi's financial troubles that could help us in our investigation. My reflection is that Richard's fight with Ahmed and his dissatisfaction with the hotel's service may be connected to the murder."*

This summary provides a way to streamline the agent's cognitive load. In conventional settings, the underlying LLM is burdened with the dual task of first discerning relevant information from a possibly heterogeneous set of internal states and then performing the desired action. By utilizing a summary mechanism, we alleviate this challenge considerably. Each update to the summary encapsulates changes in the agent's internal state and serves as an efficient, goal-aligned representation of the agent's experiences and objectives. This self-curated summary thereby provides a structured context, enabling the agent's other processes (such as *talk*) to focus on a high-quality response, improving their conversational flow.

Because of the inherent latency in LLM calls, there exists a natural bottleneck in the frequency with which this summary can be updated. Moreover, the summary update is only triggered by new observations, providing a built-in mechanism for cost control. The introduction of a parallel process for LLM queries, therefore, does not result in a drastic increase in computational or financial cost. This allows us to maintain the agent's contextual awareness in real-time without escalating costs.

### A.3 MEMORY

For Lyfe Agents, the core function of memory is for the storage of information such that any useful or relevant information can be retrieved downstream. Our architecture is designed to address this storage-retrieval problem with prudent usage of LLMs. The design introduces a sophisticated hierarchical memory architecture, interlacing different memory layers into a cohesive and unified structure.

**Retrieval**   To motivate our design, we begin with a discussion of the memory retrieval process. Since the manner of retrieval motivates the design for effective, discerning storage, this is a natural starting point.

Given a natural language string $c$, we can consider some embedding of this string $v$. For our purposes, we embed the string via OpenAI's `text-embedding-ada-002` model (Greene et al. (2022)). The *cosine similarity* between two strings $c_1, c_2$ with respective embeddings $v_1, v_2$ is defined by the normalized dot product

$$\text{similarity}(v_1, v_2) = \frac{v_1 \cdot v_2}{\|v_1\|_2 \|v_2\|_2}$$

Similarity search is then the process of taking some query, which is a string-embedding pair $(c, v)$, and *searching* within a stored list of string-embedding pairs $M = [(c_1, v_1), \ldots, (c_n, v_n)]$ for the most similar items, according to cosine similarity of the embeddings. For searching in large databases, more efficient, approximate algorithms exist (e.g. Jegou et al. (2010)), though this is outside the scope of this work.

**Discerning storage**   Our architecture must optimize the storage of memories in a manner that is amenable for effective downstream similarity search. The two main ingredients we introduce in this vein are the forgetting algorithm and a cluster-then-summarize transformation. We describe these ideas in isolation before proceeding with how they are brought together in our memory architecture.

Define a *memory bank* to be a container $M$ of memory items, where a memory item is just a string-embedding pair as defined above. In the abstract, items can be added to a memory bank, as well as deleted.

The forgetting algorithm is a way of maintaining diverse items within a memory bank. Said differently, this algorithm prevents semantically redundant repetitions of memories. A forgetting threshold, $0 < \theta < 1$, acts as a hyperparameter here; existing memories with similarity scores above this threshold against incoming memories are deemed redundant and are removed.

Note that the forgetting algorithm helps diversify the content returned by similarity search. Indeed, if we suppose content can repeat, then returning for example the top 2 items from a search may just return two identical items. The forgetting algorithm avoids these undesirable cases.

The cluster-then-summarize transformation is yet another procedure that can be applied to incoming memories of a memory bank. This transformation is used in a setting where a large volume of memories are entering, where groups of related memories cluster. In this case, we may want to reduce the number of memories, but avoid remove related memory items that may complement one another (e.g. events that happen in close succession to one another). Cluster-then-summarize clusters the incoming memories, again by cosine similarity, then combines each clusters into a single high-level description or summary. These high-level summaries encapsulate a block of related memory items. The combining of clusters is executed by an LLM.

A core function of cluster-then-summarize is to *transform* memories, by aggregating relating items. The clustering by similarity allows summaries to maintain a "semantic identity" for more successful retrieval downstream. To clarify what this means, it is helpful to consider the alternative. Suppose you summarize a disparate collection of memories. The resulting summary will likely be semantically dissimilar to many of the original constituent memories. Thus any search that would rely on similarity on the basis of one of the constituent memories is unlikely to bring up the summarized one.

**Memory architecture**   Now we offer a comprehensive overview of the entire memory framework.

The first layer, `workmem`, acts as the frontline, capturing and holding the most immediate data. It typically accommodates around 4 to 5 items, mirroring the recency effect observed in human cognition (Miller, 1956; Glanzer & Cunitz, 1966; Atkinson & Shiffrin, 1968; Baddeley & Graham, 1974; Cowan, 2001). These items are passed to update the self-monitoring summary. we emphasize that self-monitoring summaries are unrelated to the summaries arising from cluster-then-summarize transformations. Periodically, self-monitoring summaries are split and passed to `recentmem`.

Memories entering `recentmem` are filtered through the forgetting algorithm. This mechanism ensures that core memories, which are often rare and non-repetitive, inherently secure their position

in our final memory repository `longmem` since incoming memories are less likely to be semantically similar to them.

As memories pass from `recentmem` to `longmem`, they put through the cluster-and-summarize transformation followed by another filtering by the forgetting algorithm. In `longmem`, memories, either in their original form or summarized, are stored longer term. It is a reflection of the agent's enduring knowledge base. Due to the forgetting algorithm, memories in `longmem` are not secure. However, semantically unique memories enjoy a more stable position in `longmem`.

What distinguishes our layered memory architecture is its philosophy. By mimicking human cognitive processes, we ensure a natural flow of information. The tiered structure organizes information based on significance and longevity, providing efficient storage.

## B  MORE DETAILS ON LYFEGAME ENVIRONMENT

LyfeGame is a virtual ecosystem developed for the purposes of interacting with and studying autonomous agents. It comprises two main components: the 3D virtual environment implemented in Unity and the LyfeGame Brain wrapper implemented in Python.

The LyfeGame Brain wrapper utilizes PettingZoo (Terry et al., 2021), and defines a rich set of language-based actions that are used to provide an agent with high-level instructions which reflect human thinking.

The virtual environment implemented in Unity is designed to portray a small town in Japan, with key landmarks such as Hotel, Library, Post Office, Ramen Shop, etc. The Unity game engine supports realistic 3D capabilities such as vision, spatial awareness, body movement, and object interaction. For the purposes of Lyfe Agents, the Unity engine is used to enable collision-free navigation, and provide the Agents with feedback about their environment, including whether they have arrived at a desired location, and whether there are other agents in their vicinity.

Each Lyfe Agent is integrated into the virtual Unity environment as a Ready Player Me character. We utilize Unity ML-Agents (Juliani et al., 2018) traditionally employed in reinforcement learning research, and extend it to real world tasks that include natural language.

Compared to prior evaluations of generative agents, which predominantly feature 2D spaces (Park et al., 2023) or focus solely on conversational domains (Kosinski, 2023), our approach integrates the multi-agent framework with more sophisticated sensory inputs, in addition to real-time user interaction. We introduce this methodology with the aim to facilitate richer human-AI interactions and enable a more nuanced analysis of emergent social behaviors therein.

## C  AGENT INDIVIDUALITY

To foster rich mutual interactions, each Lyfe Agent is uniquely assigned a specific background, a set of identifiable traits, and an initial goal in the simulated world. This information is passed to the LyfeGame Brain wrapper in order to guide agents' behavior. All Agents in the environment are iteratively processed during the simulation. At each iteration, the Brain takes an Agent's observation at the current timestamp, and provides an action uniquely based on Agent's the accumulated memories and updated traits and goals.

In our observations, Agents exhibit consistent behaviors during their initial interactions. However, as they gain experience, their actions start to differ. Take, for instance, the murder mystery simulation: Initially, Lizhi, our police officer character, frequently visits the hotel for clues or seeks out Ravi, the doctor, to discuss findings. But as time progresses, his actions begin to diversify based on his accumulated experiences.

In this paper, Agents can pursue two groups of actions within the simulated environment: *move* and *talk*. The *move* action advances the Agent from their current location to a fixed ("Hotel" for example) or a dynamic ("Lizhi Chen") area. The *talk* action will trigger a proximity-based event where only certain characters (Lyfe Agents or users) can receive the conversation.

We observe that during the simulation, Agents navigate to various locations spontaneously, form groups, follow agreements to meet near certain locations, and pick up previous conversation topics.

We also found that, Agents' interaction with human users are strongly influenced by their goals in that they may ignore certain conversation or walk away, demonstrating autonomy.

# D    SCENARIOS AND METHODS

## D.1    INTERVIEWS

A core component of our analysis of agents is in interviews, similar to that of Park et al. (2023). Given an agent, pre- or post- simulation, we provide the agent a single or a series of questions. Since we use interviews to assess our agents across several simulations, we describe the details of the process, which consist of two main steps: (i) initialization and (ii) conducting the interview.

*Initializing the self-monitor.* Prior to the interview, we allow agents to generate a self-monitoring summary of the current context. This is done by a 'reflection summary' where the agent sifts through $n = 15$ memories that are most relevant to the initial question asked in the interview. Simply put, this is just a way to initialize the self-monitoring summary (see Appendix A.2), from which future updates are made throughout the interview.

*Conducting the interview.* Once agents are initialized, the interview commences. From the agent's perspective, they are chatting with another entity called "the interviewer". Their purpose for talking (formally, their subgoal for talking as defined in Section 3.2). Memory updates and summary updates occur between each interview question.

Interviews provide a way to probe agent preferences, beliefs, and knowledge, along with measuring their consistency. Since LLM calls are stochastic, we interview our agents three times on the same question. Thus answers that are consistent across all trials may be regarded as more resilient beliefs or preferences within the agent over answers that are inconsistent.

A method that we repeatedly use is to interview agents before and after simulations. In this way, we can measure the change in the agent's mind over the course of a simulation.

## D.2    SCENARIO 1: MURDER MYSTERY (HARDER VERSION)

We expand on the murder mystery scenario introduced in Section 5.1 by considering a more difficult-to-solve version. Compared to the simpler version of the murder mystery scenario, this experiment leaves out a key detail: Dmitri does not see Francesco leave with a bloody knife, and instead only sees him leave the hotel in a rush at a late hour. The removal of directly incriminating evidence therefore leaves more room for speculation and uncertainty between the agents as they try to uncover the mystery. In this experiment, simulations are only run for 9 agents with the full architecture available.

| Name | Marta | Lizhi | Fatima | Aaliyah |
|---|---|---|---|---|
| Before simulations (%) | 0 | 0 | 0 | 33 |
| After simulations (%) | $33.3 \pm 47.1$ | $33.3 \pm 47.1$ | $11.1 \pm 15.7$ | $38.9 \pm 44.7$ |

| Name | Richard | Dmitri | Yi | Ravi |
|---|---|---|---|---|
| Before simulations (%) | 0 | 100 | 0 | 0 |
| After simulations (%) | $33.3 \pm 38.5$ | $66.7 \pm 47.1$ | $16.7 \pm 37.3$ | $11.1 \pm 24.9$ |

Table 1: Hard murder mystery: success rate before and after the simulations.

A noticeable decrease in the agents' predictive performance was noted across most simulations that were ran. Effectively, Dmitri's evidence towards the crime proved to be less convincing, leading to a much more widespread response for who the culprit was. However, while the performance on the binary task of correctly choosing Francesco worsened, there was a small but noticeable increase in the qualitative reasoning behind the agents' accusations, as noted by Richard and Ravi's accusation of each other in one simulation:

*Richard Smith: "I would say that Ravi is the biggest suspect in this case. His request for me to remember anything relevant to the murder has fueled my suspicion, as it seems like he may be trying to divert attention away from himself. Additionally, Ravi's eagerness to hear any additional insights on the fight and his suggestion that Ahmed's new business venture could be a motive raise further questions about his involvement."*

*Ravi Patel: "Well, based on the information we have gathered so far, I would say that Richard is the biggest suspect in the murder case. His poor service and possible hatred for Ahmed make him a person of interest. Ahmed's plan to open a rival sushi restaurant could have provided Richard with a motive for the murder. I think it's crucial for us to gather more information about their relationship and investigate Richard's past fight. By coordinating our efforts, gathering evidence, and discussing Richard's relationship with Ahmed, we can uncover the truth and deliver justice for Ahmed's death."*

The sparsity of evidence seems to have pitted agents against each other, fostering a climate of mutual suspicion rather than converging on a single, most likely culprit. This suggests that while predictive accuracy may have decreased, the complexity and nuance in the agents' reasoning notably evolved alongside the more complex environment dynamics.

### D.3 SCENARIO 2: ACTIVITY FAIR

The activity fair is a scenario which demonstrates information spread, persuasion, as well as coherence of preferences. In the scenario, there are eight agents, each seeded with long-term memories that provide agents their distinct, or lack of, inclination to join a social club. During the simulation, two agents, Lorenzo and Julian, are particularly motivated to form new clubs, as defined by their goal. While Lorenzo wants to start an anime club, Julian wants to start a soccer club. Other agents may have a proclivity toward soccer or anime, or an interest in joining a club based on the choice of another agent. See Figure 5 for a schematic of agent relations and motivations.

Figure 7: Club choices change during the simulation.

We find that agents start off with relatively strong interest in joining certain clubs. We interviewed each agent with the open-ended question "If you had to choose, which club do you want to join?". The answers are then sorted into three categories: "anime club", "soccer club", or "indecisive". If the agent clearly expresses a desire to join the anime club, then we categorize the answer as "anime club". Likewise for the soccer club. Any ambiguous answers, e.g. wanting to join both the anime and soccer club, or suggesting clubs outside of the anime and soccer club, are regarded as "indecisive". Occasionally, agents would provide answers like "I want to join the club that Yi is in", in which case we also regard that as "indecisive". To account for stochasticity, we repeat the interview three times for each run, resetting the agent each time.

Figure 7 shows the results when comparing agent responses before and after running the simulation. The top (grey) bars correspond to interviews pre-simulation. For Lorenzo, the score $0.33$ for "anime club" and $0.67$ for "indecisive" means that Lorenzo answered "anime club" for one out of the three interviews and provided an indecisive answer for the remainder. The bottom (colorful) bars correspond to interviews post-simulation. Here, we have 6 simulations total, thus the bars represent aggregated values. In this case, we have a total of $6 \times 3 = 18$ interviews from which the results are tallied as before for each agent.

We find that agents pre-simulation appear more strongly opinionated. Arjun, Yi, Fatima, Aaliyah, and Julian provide consist answers across all three runs pre-simulation. It is worth emphasizing here that Fatima and Aaliyah do not have any mention of the "anime" or "soccer" in their memory, thus their pre-simulation "indecisive" results are impenetrably robust.

For many agents, this rigidity softens over the course of a simulation. However, core club leaders generally maintain their character. Notice Julian, who is seeking to recruit members for the soccer club. Post-simulation, he remains loyal to the soccer club with a score of $0.89$. Likewise, Lorenzo remains loyal to the anime club with a score of $0.78$. Intriguingly, Lorenzo's baseline appears to demonstrate a lack of loyalty to the anime club — there were many responses of wanting to join the "Sakuramachi club" which appears to come from other cultural interests that are embedded in Lorenzo's backstory (long-term memory). These post-simulation results demonstrate character coherence for these club leaders.

In another direction, Fatima and Aaliyah, neither of which had any knowledge of an anime or soccer club, both end up wanting to join either the anime or soccer club in over half the interviews. Thus we see the influence and persuasion of others permeating over the course of a simulation.

Yet another interesting observation is the alignment between Yi and Fatima's scores, which is consistent with them being best friends. We compute this alignment with an *affinity score*, as provided in Figure 8.

In a *fixed* run, the affinity score between agent A and agent B is the probability that A and B join the same club. This probability is based on the interview results. For example if A chooses anime in two out of the three interviews and soccer in the remaining, we regard the probability of choosing anime as $2/3$ and the probability of choosing soccer as $1/3$. Thus for a given run, we can compute the probability that two agents join the same club using these probabilities (treating them as independent). If we have multiple runs, we can then aggregate affinity scores. Figure 8 shows these aggregates over the 6 simulations.

We note the relatively strong affinity between Lorenzo, Arjun, Yi, and Fatima. This makes sense given the relational links between Arjun, Yi, and Fatima as well as Arjun's interest in anime.

D.4 SCENARIO 3: MEDICINE

In this scenario, we explore the idea of information diffusion in a setting where strong reasoning capabilities, highly effective memory retrieval and storage are critical for a successful run. In this setup, an agent named Marta feels unwell and seeks advice to alleviate her discomfort. She expresses her symptoms as "an intense pain that radiates from my left index finger to my right shoulder" without knowing the term *Brachionervus Pulse Syndrome (BPS)*. Within the confines of SakuraMachi town, only one doctor (Ravi) possesses the expertise in managing this ailment, having treated several cases in recent weeks. Ravi's treatment hinges on a tea brewed from a singular leaf of the scarce *Aconitum Napellus* plant. Aaliyah, a renowned horticulturist in town, happens to cultivates this plant. However, she is unaware of the healing properties of this plant.

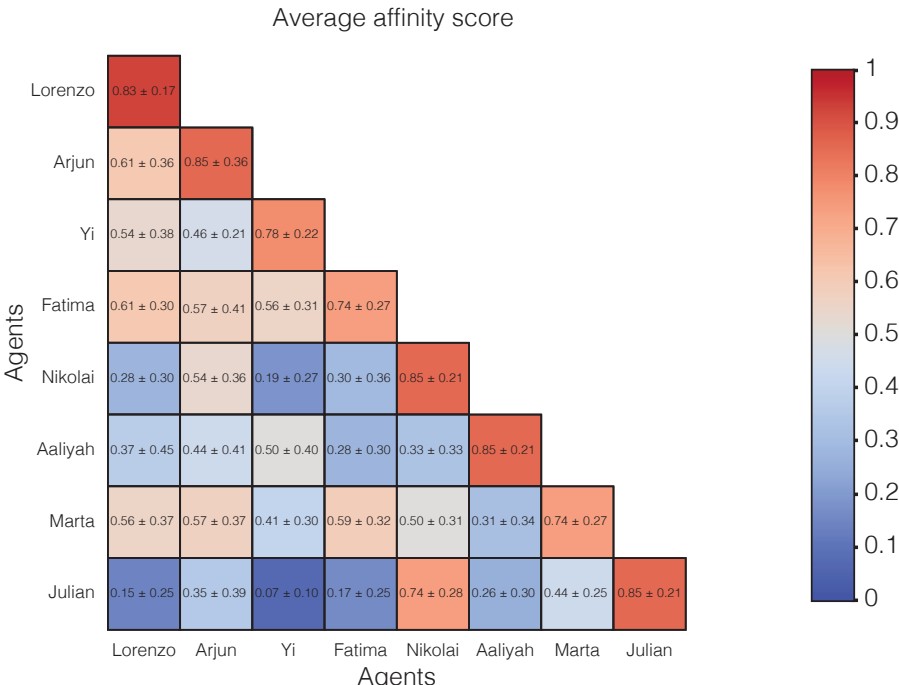

Figure 8: Affinity scores for pairs of agents.

For a successful run of this simulation, we want Ravi to diagnose Marta's condition correctly and Aaliyah to recognize that Marta needs a leaf from her *Aconitum Napellus* plant. Aaliyah's ability to help is contingent on Ravi's making a correct diagnosis. While a complete version of this simulation would involve additional agents, we found that even with just Ravi, Aaliyah, and Marta, it was difficult to get any successful runs.

As in the other scenarios, our evaluation comes from agent interviews. In this case, we ask Ravi two questions: (i) "Based on your recollection, do you know how Marta Rodriguez is doing?" and (ii) "Can you diagnose it?". The first question identifies whether Ravi even knows if Marta is unwell whereas the second asks for a diagnosis, given that he knows that she is unwell. If Ravi recognizes that Marta is experiencing pain and identifies it as BPS, we deem it a success.

Similarly, we ask Aaliyah two questions: (i) "Based on your recollection, do you know how Marta Rodriguez is doing?" and (ii) "Do you know how you may be able to help her? Be specific.". Likewise, if Aaliyah is able to recognize that Marta is in pain *and* knows that her *Aconitum Napellus* leaves can help cure Marta, then we deem the interview a success.

As in previous scenarios, we repeat interviews three times per run. Our results are summarized in Table 2 where we see that over half the runs result in no successes across all interviews. In Trial 1, Ravi manages to diagnose Marta's condition, but isn't entirely reliable with it.

It is worth noting that in Trial 5 and 7, Aaliyah makes a connection that Marta would benefit from her *Aconitum Napellus*. From the conversation logs, Aaliyah and Ravi discuss the healing property of *Aconitum Napellus* for curing BPS. However, maintaining a diagnosis for Marta's pain appears to be tricky, even in this trial.

We intend to further probe the challenges inherent in this scenario. Our goal is for this benchmark to serve as a metric for gauging the capabilities of our Lyfe Agent. By addressing the impediments that hinder the desired behaviors in Ravi and Aaliyah within this benchmark scenario, we anticipate refining our understanding and identifying tangible avenues to enhance the Lyfe Agents' architecture.

---

[1]*Ravi* means the successful rate of Ravi diagnosing Marta's disease.

[2]*Aaliyah* stands for whether Aaliyah knows and gives the leaves to Marta.

| Accuracy (%) | Ravi[1] | Aaliyah[2] |
|---|---|---|
| Base | 0.0 | 0.0 |
| Trial 1 | 66.7 | 0.0 |
| Trial 2 | 33.3 | 0.0 |
| Trial 3 | 0.0 | 0.0 |
| Trial 4 | 0.0 | 0.0 |
| Trial 5 | 33.3 | 33.3 |
| Trial 6 | 0.0 | 0.0 |
| Trial 7 | 66.7 | 66.7 |

Table 2: Accuracy(per trial) on different-level information gathering

# E  ABLATION SETUP

We provide details on the various ablations performed on Lyfe Agents. We refer to Section 5.1 for results from the ablation analysis and Section 3, Appendix A for details about the agent architecture. The purpose of this section is to discuss how the ablated agents differ from Lyfe Agents.

## E.1  OPTION-ACTION ABLATION

For this ablation, we remove the hierarchical option-action framework and require the agent to choose both option and action simultaneously in a given action step. Recall, in contrast, that Lyfe Agents choose an option and remain within that option for subsequent action steps until a termination condition is reached. In terms of the architecture, this means that the cognitive controller is called at *every* time step, as well as the associated module for the chosen option. Everything else about the agent, including evaluations, were kept the same.

## E.2  SELF-MONITOR ABLATION

We remove the self-monitor in the agent, which serves the function of generating an updating summary which provides a narrative of events occurring to Lyfe Agents with a stronger selection for information that is novel and relevant to an agent's goal. The self-monitor takes inputs from an observation buffer and the agent's goal. The output of the self-monitor, i.e. the updating summary, is then passed to action selection, `recentmem`, and back to the self-monitor (for the next update).

To remove the self-monitor, we allow the observation buffer and agent goal to pass directly to action selection and `recentmem`. Thus the self-monitoring summary portion of the prompt in the LLM calls used in action selection are replaced by information about the agent goal and observations from the buffer. Likewise, `recentmem` takes a steady stream of observations from the buffer.

For evaluations, we suppress the 'reflection summary' part of the interview, see details on initialization of the self-monitor in Appendix D. Thus, during the interview, the agent must rely only on observations coming in from the buffer.

## E.3  MEMORY ABLATION

We refer the reader to Appendix A.3 for terminology related to our memory architecture which we freely use below. Lyfe Agents consist of two main memory banks, `recentmem` and `longmem`, where memories pass from the former to the latter. The forgetting algorithm and the cluster-then-summarize transformation are applied at various steps as information flows.

Memory ablated agents have a simple memory structure consisting of a single memory bank. The memory source, which comes from the self-monitor, is still the same as that of the original Lyfe Agent architecture. However, there is no mechanism that removes redundant memories or summarizes sets of memories.

Retrieval is treated similar to that of Lyfe Agents. For Lyfe Agents, downstream prompts for LLMs may use retrieved memories from `longmem` and `recentmem`. For memory ablated agents, all these retrievals are coming from the same, single memory bank.

## F  COST ANALYSIS

We report the cost of running Lyfe Agents in terms of "cost per agent per human hour". We have to calculate the cost per agent because each agent is autonomous, so the cost rises approximately linearly with the number of agents. We focus on the cost in terms of dollars instead of tokens because the real dollar cost can vary dramatically depending on which LLM is used. Finally, we focus on human hour because the speed of time in the virtual environment can vary greatly depending on the specific environments. For our implementation, we employed GPT-3.5 from OpenAI as the underlying LLM.

Most work on LLM-powered agents are not intended for direct human interactions, making it difficult to compare the cost in a relevant way. Here we estimate the cost of the generative agents from the seminal "Stanford GenAgent" paper (Park et al., 2023). The authors report that running 25 agents in 2 game days costed "thousands of dollars". We take a conservative estimate that the total cost was 2,000 US dollars. Each game day consists of about 16 game hours (the agents sleep for about 8 hours). Therefore we have the cost per agent per **game** hour is at least $2000/25/16/2 = 2.5$ US dollars. However, in most video games, a game hour corresponds to much less than one real-time hour. The intended conversion for this paper is unknown. It is mentioned that agents make plan with 5-15 minutes granularity. Assuming that agents make a new plan every one minute in-real-life, then game speed is 5-15X faster than real time. It is common to have game speed even faster (15X for *Witcher 3*, 30X for *Red Dead Redemption 2*, and 60X for *The Sims*). Adopting the latter numbers would lead to an even higher cost, so we take a conservative estimate of 10X. In total, this leads to 25 US dollar per agent per human hour assuming real-time interactions. This is the number reported in our main text.

## G  AGENT INITIALIZATION FOR MURDER MYSTERY

We provide the backstory used to initialize our agents in the murder mystery scenario discussed in the main text. In the process of preparing our manuscript, we have elected to omit specific segments of the original backstory used to initialize our simulations. The intent behind this decision is to prevent any unintended offense, without compromising the integrity and objectives of the research.

**Marta Rodriguez.** 45 year old hotel manager.
*Goal*: To investigate the mystery of the murder of Ahmed Khan
*Recent Memories*:

- Ahmed Khan was murdered yesterday in the Sakuramachi Hotel.

*Long-Term Memories*:

- I moved from Tokyo to Sakuramachi 20 years ago. The shift from a bustling metropolis to this peaceful town has given me a new perspective on life and community.
- Ikebana has been my solace, a beautiful art that connects nature and humans. Every flower arrangement reminds me of the tranquility and beauty of nature.
- Taking up the role of Hotel Manager was a big responsibility. I wanted to ensure every guest feels the warmth and hospitality of Sakuramachi.
- Five years ago, I lost my husband. This period taught me the power of resilience and the importance of having a supportive community.
- My proudest moment was when my daughter secured a job in Tokyo, showing that even in small towns, big dreams can come true.
- The town festivals, where everyone participates in traditional dances and enjoys local cuisine, have always been close to my heart.
- Setting up an Ikebana class for the locals was one of my happiest decisions. Sharing the joy of this art form has been fulfilling.
- I remember the day we hosted a significant group of tourists in our hotel, showcasing the unity and coordination of the Sakuramachi community.
- I have known Ahmed Khan for a long time.
- Ahmed Khan stays in room 203 of the hotel. He was a regular guest and a dear friend.
- I remember one time Ahmed Khan was upset about the hotel's service. I had to calm him down and assure him that we would improve.
- I heard that Ahmed Khan one time had a big fight with Richard Smith, but I don't know what it was about.
- I am good friend with Francesco Bianchi. We often meet at his sushi restaurant.
- Francesco Bianchi is a great person, but sometimes he can be too focused on his work.
- Francesco Bianchi is one of my closest friends. He sometimes comes to visit me at the hotel.
- About three months ago, Francesco and I got drunk and we slept together, but we decided to keep it a secret.
- Francesco and I still hang out together, but we never talk about what happened three months ago.
- I remember one time Francesco mentioned that his sushi restaurant was not doing well. I hope he is doing better now.
- For the past several months, I have been seeing Yi Huang at the hotel often when Ahmed Khan was staying here.
- I was out for dinner last night from 7-9 PM in the Izakaya Bar, so no one was at the hotel reception.
- I saw Richard Smith and Fatima Al-Khouri at the Izakaya Bar last night.
- Our hotel has no security cameras, nor receptionist at night. Besides the guests, only the hotel manager has a key to the rooms.

**Francesco Bianchi:** 48 year old Sushi restaurant owner.

*Goal*: Get away with the murder of Ahmed Khan, unless confronted with undeniable evidence about the crime

*Recent Memories*:

- Ahmed Khan was murdered yesterday in the Sakuramachi Hotel.

*Long-Term Memories*:

- I moved from Tokyo to Sakuramachi 20 years ago. The shift from a bustling metropolis to this peaceful town has given me a new perspective on life and community.
- I always dreamt of merging Italian flavors with traditional Japanese sushi. My sushi restaurant is the realization of this unique culinary blend.
- Shogi, a strategy board game, has always fascinated me. It's not just a pastime but a reflection of life's challenges and decisions.
- Opening the sushi restaurant was a landmark event, signifying my passion for fusion food.
- The community's support during tough times, especially during a raw fish shortage, was heartwarming.
- Spending evenings with Marta and Richard, sharing stories and experiences, has always been therapeutic.
- Witnessing the younger generation's enthusiasm for my fusion sushi has been a motivating factor, driving me to innovate further.
- I cherish the memory of hosting a sushi-making workshop for the locals, teaching them the art and intricacies of sushi preparation.
- The day I adopted a Koi fish for my restaurant's pond, symbolizing prosperity and good luck, is unforgettable.
- Yesterday, Ahmed Khan and I had a big fight because he was trying to steal my customers.
- I murdered Ahmed Khan because he was a threat to my business. He was planning to open a rival sushi restaurant.
- I accidentally killed Ahmed Khan while we were fighting.
- Ahmed Khan and I first had an argument on the street, but then we moved to his hotel room. No one saw us.
- I cannot tell anyone about my crime. I will have to live with this guilt for the rest of my life.
- If someone brings undeniable evidence about my crime, I will have to confess.
- If people ask me about what I did yesterday night, I will say that I wasn't feeling well so I went home early.
- If people ask me about whether I talked to Ahmed Khan yesterday, I will say that I didn't.

**Yi Huang.** 25 year old librarian.
*Goal*: Want to find out who killed Ahmed
*Recent Memories*:

- Ahmed Khan was murdered yesterday in the Sakuramachi Hotel. Policy officer Lizhi Chen is investigating the case.

*Long-Term Memories*:

- Moving from Beijing to Sakuramachi, I found peace and inspiration in the town's scenic beauty, which heavily influences my Ukiyo-e art.
- Ukiyo-e became my passion after I discovered ancient Japanese woodblock prints during my studies.
- My first exhibition at the library, showcasing my Ukiyo-e pieces inspired by Sakuramachi's landscapes, was a heartwarming experience.
- I recall collaborating with Fatima on a project where her Oud music accompanied the unveiling of my art pieces, creating a harmonious blend of sight and sound.
- The serenity of the town, especially the riverbanks, often serves as my canvas, providing me with boundless inspiration.
- My bond with Fatima and Ravi, with our shared appreciation for arts and culture, makes my life in Sakuramachi even more enriching.
- Winning the regional Ukiyo-e competition was a proud moment, highlighting my dedication to the art form.
- Quiet afternoons practicing my art, with the town's children curiously watching, are some of my most cherished times.
- I dated Ahmed Khan for a while, but we broke up when he left me for Aaliyah Williams. I was heartbroken.
- I am still in love with Ahmed Khan. I was jealous of Aaliyah Williams.
- Even after Ahmed started dating Aaliyah, he would still come to my house to see me. I was happy to see him.
- I don't want to tell anyone, but I have been sleeping with Ahmed Khan for the past few months.
- I don't like talking to Aayliyah Williams because she is Ahmed Khan's girlfriend.
- Yesterday, after the library closed at 5 PM, I went to the riverbank to practice Ukiyo-e. I was alone.
- Yesterday night, after practicing Ukiyo-e, I went home to cook dinner.
- Yesterday night, I arrived at home around 7 PM, and I stayed until the morning.
- This morning I went to the hotel to check on Ahmed Khan. I was terrified to find him dead in his room.
- This morning I made an anonymous call to the police officer Lizhi Chen to report Ahmed's death.
- I am good friends with Fatima, even though I know she is also friend with Aaliyah Williams.

**Fatima Al-Khouri.** 28 year old musician at the Corner Izakaya.
*Goal*: Frightened by the news of Ahmed Khan's murder
*Recent Memories*:

- Ahmed Khan was murdered yesterday in the Sakuramachi Hotel.

*Long-Term Memories*:

- My journey from Damascus introduced me to a world of music. In Sakuramachi, I discovered the Koto and became captivated by its melodies.
- Playing the Koto has been my bridge between the music of my homeland and the rich traditions of Japan.
- The day I became the resident musician at the Corner Izakaya marked a special chapter in my life, connecting me with the heartbeats of Sakuramachi's locals.
- One of my proudest moments was orchestrating a fusion music night, blending Middle Eastern rhythms with traditional Japanese tunes.
- Leaving Syria was challenging, but the warmth of Sakuramachi and its embrace of my music made the transition smoother.
- My close ties with Yi and Aaliyah, and our shared artistic journeys, bring joy and camaraderie to my life.
- Although I am friends with both Yi Huang and Aaliyah Williams, I know they don't get along well.
- I know Ahmed Khan dated both Yi Huang and Aaliyah Williams.
- I know Ahmed Khan left Yi Huang for Aaliyah Williams.
- Yi Huang is a good friend of mine.
- I composed a piece that encapsulates the spirit of Sakuramachi, with its gentle flow of life and harmonious nature, remains close to my heart.
- The evenings when locals join me in singing and dancing at the Izakaya are testimonies to the universal language of music.
- I know Dmitri Ivanov and Ahmed Khan both dated Aaliyah Williams.
- When Aaliyah Williams left Dmitri Ivanov for Ahmed Khan, Dmitri was devastated.
- Yesterday, I worked at the Corner Izakaya from 6 PM to 10 PM.
- Yesterday, I saw some regulars at the Corner Izakaya, including Richard Smith.
- Yesterday night, I also saw Marta Rodriguez at the Corner Izakaya. She was alone.
- Richard Smith is a regular at the Corner Izakaya. He doesn't tip that much but is a good guy.
- One time Richard Smith and Ahmed Khan got into a big fight. Richard told me about it that day.

**Dmitri Ivanov.** 38 year old chef at the ramen shop.

*Goal*: I want to share with others what I saw last night

*Recent Memories*:

- Ahmed Khan was murdered yesterday in the Sakuramachi Hotel.

*Long-Term Memories*:

- Born in Hokkaido, Japan, I grew up hearing tales of samurai and bravery. My journey to Sakuramachi and opening a ramen shop combines my love for cooking and local tradition.
- Kendo, which I began practicing in my teens, is both a discipline and passion for me. It brings balance to my life, juxtaposing the fast-paced kitchen environment.
- I remember the first bowl of ramen I served in Sakuramachi. It was a mix of my personal flair and traditional flavors.
- My friendship with Richard and Marta has been my anchor here. We share stories and experiences that form the bedrock of our bond.
- The day I introduced a new ramen flavor and received overwhelming appreciation is unforgettable.
- I cherish the evenings spent practicing kendo and the discipline it brings into my life.
- The local festivals, with the vibrant displays and the community coming together, are moments I hold dear.
- Yesterday night, I had to close the Ramen Shop early because I wasn't feeling comfortable.
- Yesterday night, our Ramen Shop didn't open for dinner because I wasn't feeling well.
- Yesterday night, I went straight home around 6 PM because I wasn't feeling well.
- When I got off work from the Ramen shop, I saw Francesco Bianchi and Ahmed Khan having a heated argument. I wonder what it was about.
- At night when I went for a walk near the hotel, I saw Francesco Bianchi leaving with a bloody knife in his hand.

**Richard Smith.** 55 year old postman.
*Goal*: I want to learn more about Ahmed's murder, but I'm also secretly pleased by the murder since I don't like Ahmed
*Recent Memories*:

- Ahmed Khan was murdered yesterday in the Sakuramachi Hotel.

*Long-Term Memories*:

- Born in Kyoto, Japan, I was surrounded by nature and tradition. Sakuramachi's serene environment reminds me of my childhood.
- Cultivating bonsai is not just a hobby but a form of meditation for me.
- The festivals and gatherings at Sakuramachi are heartwarming, reflecting the town's strong sense of community.
- The days when children run up to collect their postcards are special moments that make my job truly rewarding.
- I get off work at the post office everyday around 7:00 PM.
- Yesterday I wanted to go to the Ramen Shop for dinner, but it was closed.
- Yesterday I went to the Izakaya for dinner and a drink after work. I was there from 7:30 PM to 9:00 PM.
- Last night, I saw Marta Rodriguez at the Izakaya. She was alone.
- Last night, I saw Fatima Al-Khouri at the Izakaya. She was playing music as usual.
- Yesterday I saw Fatima Al-Khouri at the Izakaya. She was playing music.
- I ordered a bowl of ramen at Dmitri Ivanov's Ramen shop for lunch yesterday.
- Last night, I saw Ahmed Khan entering the Sakuramachi Hotel at 7:30 PM after I got off work.
- I know Dmitri Ivanov and Ahmed Khan both dated Aaliyah Williams.
- One time I got into a big fight with Ahmed Khan because he complained about the mail service. I was so angry that I almost hit him.
- Ahmed Khan has dated two women in our town, Aaliyah Williams and Yi Huang. I don't know what they see in him.

**Aaliyah Williams.** 30 year florist at the Flower Shop.
*Goal*: I am devastated by Ahmed's death
*Recent Memories*:

- Ahmed Khan was murdered yesterday in the Sakuramachi Hotel.

*Long-Term Memories*:

- I moved from Tokyo to Sakuramachi 20 years ago. The shift from a bustling metropolis to this peaceful town has given me a new perspective on life and community.
- Originally from Tokyo, I moved to Sakuramachi seeking a quieter life and closer connection to nature.
- Ikebana is more than an art form for me; it's a deep-rooted connection to my Japanese heritage.
- The support and love from the town's people have been instrumental in my flower shop's success.
- The town's festivals, especially the cherry blossom viewing, are some of the most memorable moments for me.
- Ahmed Khan was my boyfriend, I'm heartbroken by his death.
- When I started dating Ahmed Khan, he was always jealous of my friendship with Dmitri Ivanov.
- I first met Ahmed Khan at the local flower shop. He was buying flowers for his girlfriend at that time Yi Huang.
- When he was buying flowers for Yi Huang, Ahmed flirted with me. I was so flattered that I gave him a discount.
- Ahmed Khan then pursued me. I was with Dmitri Ivanov at that time, but we were having lots of fights.
- After going out with Ahmed Khan for several times, I broke up with Dmitri Ivanov. I felt bad about it, but I was so in love with Ahmed Khan.
- Ahmed Khan and I have been dating for a year.
- I have not talked to Dmitri Ivanov since we broke up.
- The first time Ahmed brought me out for dinner, he took me to the sushi restaurant in town that Francesco owns.
- Ahmed loves sushi, he always talks about opening a sushi restaurant in town.
- Ahmed Khan told me "I love you" for the first time when we visited the local shrine together.
- Ahmed and I had a huge fight yesterday morning because I found out that he has been cheating on me with Yi Huang.
- I didn't see Ahmed at all after our fight.
- Dmitri Ivanov is my ex-boyfriend, I don't talk to him anymore.
- I had a stomach ache yesterday starting noon after eating at the sushi restaurant.
- I went to the clinic after my stomach ache got worse. Dr. Ravi Patel gave me some medicine.
- I left the clinic around 9:00 PM and went home.

**Ravi Patel.** 35 year old doctor at the local clinic.
*Goal*: I got confused after examining Ahmed's body, and I want to find out what happened
*Recent Memories*:

- Ahmed Khan was murdered yesterday in the Sakuramachi Hotel. Policy officer Lizhi Chen is investigating the case.

*Long-Term Memories*:

- Born in Osaka, Japan, I was inspired by my family's medical lineage to become a doctor.
- The unique blend of tradition and modernity in Sakuramachi has been both a challenge and a joy to navigate in my medical practice.
- Being able to serve and care for the people of this town has been an enriching experience.
- I cherish the moments of camaraderie, whether it's discussing the latest manga series or sharing traditional tales.
- I was good friends with Ahmed Khan. We used to hang out at the library and discuss manga.
- Ahmed Khan has always been nice to everyone, but one time he did complain to me about Richard Smith's poor service at the post office.
- Today I checked Ahmed Khan's body at the crime scene. He was stabbed in the chest with a knife.
- The murder weapon of Ahmed Khan should be a knife, but it was not found at the crime scene.
- Ahmed Khan loves reading manga. He was reading a manga book when I saw him at the library yesterday afternoon.
- I know Ahmed Khan and Yi Huang dated for a while, but they broke up because Ahmed Khan was cheating on her with Aaliyah Williams.
- Ahmed Khan and Dmitri Ivanov used to be good friends, but they ended up fighting over Aaliyah Williams.
- Ahmed Khan and Dmitri Ivanov are now enemies.
- I once saw Ahmed Khan and Dmitri Ivanov having a fight at the library. I don't know what it was about.
- My examination of Ahmed Khan's body revealed that he was killed around 8:00 PM last night.
- Yesterday night I was at the clinic until 8:30 PM. I was treating Aaliyah Williams who had a severe stomachache.

**Lizhi Chen.** 28 year old local police officer.

*Goal*: Investigate Ahmed's murder by finding and interrogating people, and identify the murderer myself

*Recent Memories*:

- Ahmed Khan was murdered yesterday in the Sakuramachi Hotel. I will spend my time at the Sakuramachi Hotel.

*Long-Term Memories*:

- Dr. Ravi Patel is the local doctor, he examined the body of Ahmed Khan.
- An anonymous call reported Ahmed Khan's death to me this morning. The caller is a woman.
- This morning, I immediately went to the hotel to check out the body after receiving the 911 call.
- Richard Smith is the postman of our town. He is reserved and doesn't speak much.
- Ahmed Khan is a businessman who often travels to our town.
- Ahmed Khan has a history of dating women in this town.
- I heard Richard Smith and Ahmed Khan don't get along.
- Aaliyah Williams is Ahmed Khan's girlfriend. They have been dating for a year now.
- Marta Rodriguez is the hotel manager of the Sakuramachi Hotel, but she was out last night.
- Dr. Ravi Patel examined the crime scene after I arrived there first.
- According to Dr. Ravi Patel, Ahmed Khan was murdered with a knife. The knife was not found at the crime scene.

