# OpenReview forum: "Lyfe Agents: generative agents for low-cost real-time social interactions"
_ICLR.cc/2024/Conference — Submitted to ICLR 2024_

### Official Review · Reviewer_XnQy · 2023-10-15

**Soundness:** 2 fair
**Presentation:** 3 good
**Contribution:** 1 poor
**Rating:** 3
**Confidence:** 2

**Summary:**

The paper introduces a generative agents architecture called lyfe agents. The claim is that it is more economically than baselines by using hierarchical action selection scheme. Furthermore, the architecture introduced a self-monitoring module to facilitate more coherent behavior of agents. Finally, a summary-then-forget mechanism is introduced to reduce redundant information and keep what is salient. Overall, the paper is able to produce coherent, objective driven agents, simulated in their own environment called LyfeGame.

**Strengths:**

- Tackles the problem of cost efficient generative agents and effective memory systems. The topic is very relavant to the conference and as a field. The paper is presented in a simpe manner, and the presented material is easy to understand.
- The economic value seems like a valuable topic to explore, since agents should not need a heavy LLM to perform low level actions, which is very expensive
- provided clear advantage over generative agents: cost, which is also significant reduction

**Weaknesses:**

- The paper uses mainly people in the field already practice, e.g., reflecting on top of reflections in generative agents is similar to the self-monitoring in lyfe agents, and time as a inverse importance metric in generative agents is similar to summary-then-forget
- paper is missing alot of key implementation details in order to reproduce, see questions section for a list of them
- the environment seem like a navigatable chatroom (only walk/talk actions), seems a bit simplified to incentivize more complex social behaviors

**Questions:**

- Could you provide more details of the
    - memory system, how was it implemented, what retrieval/embedding mechanisms were employed
    - hierarchical actions were implemented in code? or pretrained policy?
    - clustered using what technique? what was the embedding employed?

---

> ### Author Response · Authors · 2023-11-11
> **We thank the reviewer for the constructive criticism!**
>
> We believe we can address all of the reviewer’s concerns. Here we provide a preliminary response.
>
> ### **Novelty of techniques**
>
> **Q:** The paper uses mainly [techniques that] people in the field already practice, e.g., reflecting on top of reflections in generative agents is similar to the self-monitoring in lyfe agents, and time as a inverse importance metric in generative agents is similar to summary-then-forget.
>
> **A:** We believe there are **important differences** between the techniques mentioned and our proposed techniques. We are currently **conducting new experiments** to directly compare our summarize-and-forget technique with time-based importance metric. We expect to update the reviewer with our results in 3-4 days.
>
> Meanwhile, we will edit the text to clarify the difference and innovation of our techniques. We provide some of that clarification here for the reviewer’s convenience. It is important to note that Park et al.’s reflection does not correspond to our self-monitoring module and we argue that this is a *novel* contribution:
>
> *Self-Monitoring Summary*: The function of reflect in Park et al. 2023 is to derive higher-order reasoning on items that are passed to the memory stream. The output of reflection are additional items which are added to the memory stream. Our agents have a reflect action as well which performs this operation — it is different from Park et al. in that it requires a single language model call and is triggered at the end of conversations, but nonetheless serves a similar function.
> The self-monitoring summary is responsible for providing coherent context to the agent for decision making. Certainly, context *must* be provided to agents for sensible action selection, and such a mechanism is present in Park et al. However, it’s our method for providing coherent and situationally persistent context here that is novel. The method is cheap in terms of LLM calls required and conditions goal-adherence. It follows a rolling update rule, where goal-relevant recent observations and related memories modify the previous summary, in a single language model call.
> 	In contrast, the agents from Park et al. generate a context summary from their memory stream through three LLM calls. This was generated through two prompts that retrieve memories and view the queries “What is [observer]’s relationship with the [observed entity]?” and “[Observed entity] is [action status of the observed entity]?”, and their answers summarized together. Another call is used to then decide based on the context summary whether to react and what that reaction would be.
> Bringing this all together, Park et al’s method for providing a context summary requires three language model calls for every action decision step. Our approach of using a rolling update has not only the benefit of using a single LLM call for updating, but also the benefit of reusing these contexts multiple times — we don’t need to update the summary frequently since the situational context isn’t expected to change rapidly.
>
> *Summarize-then-Forget Memory*: For the comparison between time-based importance and our summarize-then-forget mechanism, as mentioned above, we will run comparisons between these architectures. However, we want to emphasize that the motivation behind summarize-then-forget was to reduce the storage of redundant memories. This is important when under the constraint of retrieving fewer memories for action selection — if you can only retrieve 4 memories, it is much less useful if these memories are repetitive.
>
>
> ### **Key implementation details missing in the main text**
> Reviewer correctly pointed out the lack of implementation details. We are adding **substantially more details** to the main text right now. We can provide the answers to the reviewer’s specific questions for the reviewer’s convenience once these changes are made.
>
> ### **Environment too simplified**
>
> **Q:** “A navigable chatroom (only walk/talk actions)...[too] simplified to incentivize more complex social behaviors”
>
> **A:** We agree that for this paper, the environment is simplified and does not support complex agent-object interactions or physical movements. We will emphasize this limitation in the Conclusion & Discussion section.
>
> We do think that a wide range of complex social behaviors can still be simulated in a navigable chatroom. A house-warming party, a presidential debate, and a diplomatic negotiation are all largely conducted through verbal communications.

---

> ### Author Response · Authors · 2023-11-22
>
> Dear Reviewer XnQy,
>
> We appreciate your insightful comments and suggestions. We would like to present updated results from our new simulations after modifications and look forward to further discussions.
>
> To make a better comparison of our cost with other existed counterparts, especially that of the architecture from Park et al., we took their memory and implemented it in our agent for comparison. We kept our self-monitoring summary and action selection modules, but replaced our memory. Our runs indicate that our memory utilizes approximately 30 times fewer tokens than that of the Park et al. memory. Due to differences in latency, we normalize the comparison by looking at the rate of token cost for the memory per action-selection. For our base architecture, the cost was 22.00 tokens per action-selection whereas for Park et al. we found 612.53 tokens per action-selection. We note that the performance using Park et al.’s architecture was also not great, though we anticipate this may be due to our manner of defining memory keywords for which there wasn’t a straightforward adaptation from Stanford to our setting.
>
> We are in the process of enhancing our paper with additional technical details and a more logical structure for explanations. To date, we have expanded the main text with essential technical information and supplemented this with detailed data in the Appendix. Our efforts to further clarify these technical aspects are ongoing.
>
>
> Sincerely
> Submission 8143

---

### Official Review · Reviewer_tWHo · 2023-10-22

**Soundness:** 2 fair
**Presentation:** 3 good
**Contribution:** 3 good
**Rating:** 5
**Confidence:** 2

**Summary:**

This paper introduces a novel generative agent architecture for real-time social interactions.

Achieving real-time responses necessitates high throughput but also incurs significant computational costs. The proposed work aims to develop a cost-effective agent with key features such as: 1) an option-action selection strategy that generates social actions hierarchically, similar to an option framework, 2) a self-monitoring module that continuously updates an activity summary for streamed observations, and 3) a summarize-and-forget memory that discards redundant previous memories based on their similarity with newer ones. By operating multiple agents with these features, social interactions can emerge.

Experimental results on two social interaction tasks (murder mystery and activity fair) validate the effectiveness of the proposed agents.

**Strengths:**

**Novelty**: The specific architecture of the generative agents developed in this work appears novel, although I am not an expert in this particular field.

**Quality**: The development of a comprehensive framework for social interactions by multi-generative agents likely required a significant engineering effort. From an engineering perspective, the quality of the proposed work is high. Each technical component of the proposed agent is thoroughly evaluated in the ablation study.

**Clarity**: The paper is mostly well-written and easy to understand. Although many technical implementation details are omitted from the main sections, this is acceptable for a conference paper submission with a page limit.

**Significance**: The demonstrated results appear significant (although I'm not certain if they truly outperform other work, as discussed in the next section.)

**Weaknesses:**

While the proposed work seems to be of high quality from an engineering perspective, its significance as a conference paper is somewhat limited in its current form, due to what I perceive as insufficient experimental evaluation.

Here are my concerns:
- Although I understand that all the components of the proposed agent contribute to task success rates, it's unclear how crucial they are for the agent to be "cost-effective", which was the original motivation of this work as stated in the introduction.
- While the proposed agent is performant, it's unclear if this performance is consistent across various base LLMs. The experiment appears to use GPT-3.5. What would happen if we use different LLMs such as Alpaca and GPT4ALL, or LLMs with various model capacities? Without this information, it remains unclear whether using GPT-3.5 is a crucial requirement or not. Also, from the perspectives of reproducibility and transparency, using open-sourced LLMs would be more desirable.
- The cost analysis was not very convincing. Currently, "cost per agent per human hour" is reported. However, for tasks with specific goals such as murder mystery, isn't it more essential to show the cost required until the task is successfully completed? Otherwise, agents with very low throughput (due to, say, insufficient compute resources) will be evaluated as low cost, which I don't believe this paper intends to argue. Furthermore, reporting costs in USD seems unhelpful. This would result in running open-sourced models in a region with cheap electricity as an optimal choice. What happens if the service provider changes API usage prices in the future? Why not just report the number of interactions or number of tokens consumed until task completions?

**Questions:**

Though I'm not an expert in this field, I found this work overall interesting. My main concerns are about whether the arguments "low-cost", "cost-effective", etc. are sufficiently supported, which I don't believe they are in the current submission.

- Can the contribution of each technical component be evaluated in terms of cost-effectiveness, such as the number of interactions or the number of tokens consumed?
- Can the base LLM be replaced from GPT-3.5 to other open-sourced LLMs such as Alpaca https://github.com/tatsu-lab/stanford_alpaca and evaluated with several model capacities?
- Can the cost analysis be performed not based on USD/agent/hour but something else that does not depend on some API's price setting? For example, could we use the number of interactions or tokens required until task completion?

---

> ### Author Response · Authors · 2023-11-11
> **Thank you for the constructive feedback!**
>
> We are currently running a number of new experimental evaluations, detailed below. We will report the results in 3-4 days. We believe we can address all concerns raised.
>
> ### **Relationship between component and cost unclear**
> **Q:** ..unclear how crucial [components] are for the agent to be "cost-effective".
>
> **A:** We will add a more **focused comparison of our components with alternatives** to the main text, and explain how these components strongly improve the performance while keeping the cost low.
>
> ### **Alternative LLMs**
> **Q:** …unclear if this performance is consistent across various base LLMs
>
> **A:** We agree. We are currently **re-running our simulations** where the LLM is Llama-2-70B, Mistral-7B, or GPT4. The first two are open-sourced.
>
> ### **Better cost analysis**
> **Q:** The cost analysis was not very convincing. Currently, "cost per agent per human hour" is reported…isn't it more essential to show the cost required until the task is successfully completed?
>
> **A:** Because the task can’t always be completed (e.g. with component ablations), we can’t calculate tokens needed for task completion. We will, however, show how the performance changes as a function of **both tokens and monetary cost**.
>
> **Q:** reporting costs in USD seems unhelpful… Why not just report the number of interactions or number of tokens consumed until task completions?
>
> **A:** We will report both token consumption and monetary cost. We explain the reasoning below and will add it to the main text:
> “We report both token consumption and monetary costs. When using the same base LLM, reporting token consumption gives a fair comparison and is robust to API price change. However, for different base LLMs that may vary dramatically in model size (e.g. GPT4 vs 7B open-source models), the relative monetary cost is a (flawed) approximation of the compute resource used.”
>
> **Q:** Can the contribution of each technical component be evaluated in terms of cost-effectiveness, such as the number of interactions or the number of tokens consumed?
>
> **A:** We are rewriting our main text to better explain how our proposed modules reduce the cost while keeping the performance high. We are planning to look into adapting Stanford GenAgent’s memory and retrieval architecture into our agents as a better point of comparison, and relying on token-based cost for comparison rather than purely USD. We note that the GenAgent architecture is interwoven with the environment, so this adaption may face difficulties, but we will regardless add a **new section that explains why it’s difficult to directly compare them**. The low-cost nature of our work remains true, but we will not emphasize it by comparing it with the Stanford paper. (Stanford paper did not use GPT-4 though, they used text-davinci-3, which is still more expensive)

---

> > ### Comment · Reviewer_tWHo · 2023-11-11
> > **Thank you**
> >
> > Thank you for the response and I’m looking forward to results from the new experiments!
> >
> > For the cost analysis metric, it may be useful to extend the “success weighted by path length” that is often used in the study of embodied navigation. See: https://arxiv.org/abs/1807.06757

---

> ### Author Response · Authors · 2023-11-22
>
> Dear Reviewer tWHo,
>
> We appreciate your insightful comments and suggestions. We would like to present updated results from our new simulations after modifications and look forward to further discussions.
>
> We will continue to work on it as some of these results are preliminary. The items are (1) comparing the performance of our agents with other language models, (2) cost comparison of our agent memory vs. an implementation using Stanford’s memory, and (3) progress on the overall structure of the paper. For (1), we find that GPT3.5 and GPT4 have comparable performance whereas open-source models tend to do worse. For (2), we find that our memory uses about 30X fewer tokens per action selection. For (3), we are in the middle of changes to the structure, having already modified our literature review for better contextualization and the introduction to emphasize that our agents are intended for social interaction. Further details on these items are provided below.
>
> **1. Running simulations for other language models.**
>
> We ran our simulations using GPT4, Mistral-7B, and Llama-70B and present the results of our preliminary exploration. To normalize these comparisons, we modified the simulation length for each one so that they had approximately the same number of action selection decisions per simulation. We preface that these results are preliminary, we mention some issues below. The results of each trial:
>
> | Model   | Mean ± Standard Error |
> |---------|-----------------------|
> | gpt3.5-Turbo   | 0.58 ± 0.08            |
> | gpt4   | 0.44 ± 0.09            |
> | llama2-70B   | 0.13 ± 0.06            |
> | mistral-7B   | 0.23 ± 0.09            |
>
>
> The GPT3.5 runs include the simulations from the original submission along with some additional runs we made during the rebuttal period. We note that there is a caveat for Mistral and Llama, the percentage of incorrectly formatted outputs was higher. This led to requiring longer runs (since effective action selection takes longer to get right) and there may be other confounding factors contributing to the performance.
>
> **2. Comparison of cost with Stanford**
>
> To make a better comparison of our cost with that of the architecture from Park et al., we took their memory and implemented it in our agent for comparison. We kept our self-monitoring summary and action selection modules, but replaced our memory. Our runs indicate that our memory utilizes approximately 30 times fewer tokens than that of the Park et al. memory. Due to differences in latency, we normalize the comparison by looking at the rate of token cost for the memory per action-selection. For our base architecture, the cost was 22.00 tokens per action-selection whereas for Park et al. we found 612.53 tokens per action-selection. We note that the performance using Park et al.’s architecture was also not great, though we anticipate this may be due to our manner of defining memory keywords for which there wasn’t a straightforward adaptation from Stanford to our setting.
>
> **With your great suggestion on extending“success weighted by path length”, we are doing experiments and plan to further investigate this.**
>
> **3. Modifications to the paper**
>
> We have incorporated several changes to the paper, though this is ongoing. The introduction better emphasizes the aim of this work as developing agents for social simulation and the relation between the modules and cost effectiveness, with details in Section 3 --- more on this in the next paragraph. We moved the literature review to follow the introduction, better contextualizing our work amidst them. In particular, we discuss language model powered agents, existing benchmarks and problems therein, long-term memory via vector database, and coherent goal-directed behavior with reflection. More details will be provided on model architecture.
>
> We are still in progress of making these changes, but want to summarize the argument for the cost-effectiveness of each module. For action-selection our argument remains the same, to save calls on choosing higher level actions through the hierarchical design. We plan to incorporate our investigations from item (2) above to demonstrate our consideration for cost in memory. We also emphasize the cost-effectiveness and novelty of the self-monitoring summary as a way to provide context for action-selection. In particular, we point out that this provides context for action-selection with a slow update that allows for reuse across multiple action steps — as the contextual narrative changes slowly relative to action steps.
>
>
> Sincerely
> Submission 8143

---

### Official Review · Reviewer_4MNZ · 2023-10-30

**Soundness:** 1 poor
**Presentation:** 2 fair
**Contribution:** 1 poor
**Rating:** 3
**Confidence:** 4

**Summary:**

This paper proposes an LLM-based agent named Lyfe Agents, which is claimed to enable real-time interactions with low computational cost. The authors propose an option-action framework and a new memory mechanism for the Lyfe Agents. Lyfe Agents are claimed to reduce the number of queries by 95% compared to Generative Agents.

However, this paper lacks novelty and contributes marginally to the field of LLM-based agents. The claims of 'low-cost,' 'real-time,' and 'generative' are not sufficiently substantiated for a scientific research paper. It appears more akin to an application of existing LLM-based agents with a designed script.

**Strengths:**

1. The pipeline in Figure 2 is clear and interesting.
2. The paper is well-written and presents its ideas in a clear, concise manner.

**Weaknesses:**

This paper is difficult to regard as a research paper, or even as a technical report. It more closely resembles a manual for implementing an LLM-based agent framework in two scenarios designed by the authors.

The logic of this paper is not rigorous.

1. The authors claim that existing LLM-based agents are redundant in leveraging fewer queries to empower the agents. However, the problem they address is not convincing. The authors treat the redundancy issue as common knowledge within the community, which lacks objectivity.

2. The overall logic of this paper is unclear. What motivates the proposal of the three modules for Lyfe Agents, and how do these modules differ from those in other LLM-based agents? The improvements offered by these modules are difficult to discern.

3. The authors do not fairly compare Lyfe Agents with existing LLM-based agents. The paper's most significant claim pertains to the 'low-cost' agents depicted in Figure 6. However, the authors should attempt to keep all variables consistent when comparing the queries of LLMs. Comparing generative agents with Lyfe Agents in different scenarios is unjust! Also, the lyfe agents should compare with other general social simulation agents such as [1,2].

4. There are no objective evaluation metrics for Lyfe Agents other than cost per hour (which, as stated in point 3, is an unfair comparison). The police success rate shown in Figure 3b is absurd. It is based on a self-designed script with predetermined and clear answers, which is not convincing and should not serve as an evaluation metric in a scientific paper.

5. The proposed framework in figure 2 should be a general LLM-based agents framwork. The lyfe agent should be evaluated in other well-know scenarios such as games and generative agents [3,4].

6. Generative agents [3] propose reflection to summarize the high-level memory. What is the difference between the proposed memory mechanism and the one in generative agents [3]?


[1] Jiaju Lin, Haoran Zhao, Aochi Zhang, Yiting Wu, Huqiuyue Ping, and Qin Chen. Agentsims: An open-source sandbox for large language model evaluation. arXiv preprint arXiv:2308.04026, 2023.

[2] Lei Wang, Jingsen Zhang, Xu Chen, Yankai Lin, Ruihua Song, Wayne Xin Zhao, and Ji-Rong Wen. Recagent: A novel simulation paradigm for recommender systems. arXiv preprint arXiv:2306.02552, 2023.

[3] Joon Sung Park, Joseph C. O’Brien, Carrie J. Cai, Meredith Ringel Morris, Percy Liang, and Michael S. Bernstein. Generative agents: Interactive simulacra of human behavior. In In the 36th Annual ACM Symposium on User Interface Software and Technology (UIST ’23), UIST ’23, New York, NY, USA, 2023. Association for Computing Machinery.

[4] Guanzhi Wang, Yuqi Xie, Yunfan Jiang, Ajay Mandlekar, Chaowei Xiao, Yuke Zhu, Linxi Fan, and Anima Anandkumar. Voyager: An open-ended embodied agent with large language models. arXiv preprint arXiv:2305.16291, 2023.

**Questions:**

1. What is the problem (motivation) addressed in this paper? llm-based agents with too many queries? What is the factor leading to the "extensive queries" of existing LLM-based agents? Are the queries really extensive? Is the memory mechanism resulting in the "extensive queries"?

2. How do the proposed Lyfe agents solve the problems addressed by the existing LLM-based agents? Which module contributes the most to efficiency?

3. How are the scripts (scenario 1 & 2) driven?

4. What is the LLM used in the experiments?

5. Could existing LLM-based agents be real-time agents if the LLM is running locally?

---

> ### Author Response · Authors · 2023-11-11
> **Thank you for the comprehensive suggestions! (1/2)**
>
> ### **Are redundant LLM calls a real problem?**
> **Q:** The authors claim that existing LLM-based agents are redundant in leveraging fewer queries to empower the agents. However, the problem they address is not convincing. The authors treat the redundancy issue as common knowledge within the community, which lacks objectivity.
>
> **A:** Our understanding of the reviewer’s response is that it is not true or common knowledge that LLM-based agents may use too many queries. We agree that we had taken this assumption for granted. Our work’s focus however is on reducing these calls from the perspective of cost reduction. For a better comparison, we are investigating the use of Park et al.’s memory and retrieval architecture into our agent to demonstrate the difference in cost (evaluated by token usage). We note that the Stanford GenAgent architecture is interwoven with the environment, so this adaptation may face difficulties, but we will regardless add a **new discussion section that explains why it’s difficult to directly compare them fairly**.
>
>
> ### **Need for better logic**
> **Q:** The overall logic of this paper is unclear. What motivates the proposal of the three modules for Lyfe Agents, and how do these modules differ from those in other LLM-based agents? The improvements offered by these modules are difficult to discern.
>
> **A:** We are including substantially more technical details about our agents’ implementations, and are contextualizing our contributions by more directly comparing the modules with relevant modules in the literature.
>
>
> ### **Comparison with other LLM-based agents**
> **Q:** The authors do not fairly compare Lyfe Agents with existing LLM-based agents. The paper's most significant claim pertains to the 'low-cost' agents depicted in Figure 6. However, the authors should attempt to keep all variables consistent when comparing the queries of LLMs. Comparing generative agents with Lyfe Agents in different scenarios is unjust! Also, the lyfe agents should compare with other general social simulation agents such as [1,2].
>
> **A:** We thank the reviewer for suggesting comparisons with other agents. In the current stage of LLM-based agent research, fair comparison between agents is generally difficult because the application domains can be vastly different. We agree with this assessment. Therefore we are **adding a section** detailing other agents, describing this challenge. We are also **removing direct comparison** of the cost between Stanford Generative Agent from our main text.
>
> For the same reason, it is difficult to compare Lyfe Agents with [1, 2]. The AgentSims paper [1] proposes a Sandbox like the Stanford paper [3], but neither [1, 3] provides a standardized benchmark to evaluate agents. The Wang paper [2] is focused on a recommender framework that is different from our goal of simulating social behavior in virtual environments.
>
> That said, we are working on implementing some components of the Stanford paper [3] into our agents and evaluating them, as mentioned in our response to an earlier question. We will provide an update as soon as possible.
>
> ### **Evaluation of Lyfe Agents in other scenarios**
> **Q:** The proposed framework in figure 2 should be a general LLM-based agents framework. The lyfe agent should be evaluated in other well-known scenarios such as games and generative agents [3,4].
>
> **A:** Our work focuses on social interactions, so we intentionally avoided games and evaluations such as Minecraft [4]. It is also difficult to evaluate our agents fairly with agents such as Voyager [4] that execute actions by writing code.
>
> The Stanford paper [3] does not provide a reproducible benchmark to evaluate on. The only quantitative benchmark provided requires human participants to judge whether the agents’ answers to interview questions are reasonable. This evaluation is too expensive and time-consuming for us and most practitioners in the field.
>
> In fact, for these reasons, the more agent papers referenced by the authors [1, 2] as well as other recent agent papers (e.g. ChatDev from Tsinghua) typically don’t evaluate their agents on other scenarios.

---

> ### Author Response · Authors · 2023-11-11
> **Thank you for the comprehensive suggestions! (2/2)**
>
> ### **Novelty of technique proposed**
> **Q:** Generative agents [3] propose reflection to summarize the high-level memory. What is the difference between the proposed memory mechanism and the one in generative agents [3]?
>
> **A:** We are editing the text to better clarify how the reflection from Park et al. and our self-monitoring summary differ. Likewise for time-weighted decay and the forgetting algorithm. This textual edit and new simulations should explain the difference.
>
> We are currently **conducting new experiments** to directly compare our summarize-and-forget technique with time-based importance metric.
> Meanwhile, we will edit the text to clarify the difference and innovation of our techniques. We provide some of that clarification here for the reviewer’s convenience. It is important to note that Park et al.’s reflection does not correspond to our self-monitoring module and we argue that this is a novel contribution:
> 	The function of reflect in Park et al. 2023 is to derive higher order reasoning on items that are passed to the memory stream. The output of reflection are additional items which are added to the memory stream. Our agents have a reflect action as well which performs this operation — it is different from Park et al. in that it requires a single language model call and is triggered at the end of conversations, but nonetheless serves the same function.
> 	The self-monitoring summary is responsible for providing coherent context to the agent for decision making. Certainly, context must be provided to agents for sensible action selection, and such a mechanism is present in Park et al. However, it’s our method for providing coherent and situationally persistent context here that is novel. The method is cheap in terms of LLM calls required and conditions goal-adherence. It follows a rolling update rule, where goal-relevant recent observations and related memories modify the previous summary, in a single language model call.
> 	In contrast, the agents from Park et al. generate a context summary from their memory stream through three LLM calls. This was generated through two prompts that retrieve memories and view the queries “What is [observer]’s relationship with the [observed entity]?” and “[Observed entity] is [action status of the observed entity]?”, and their answers summarized together. Another call is used to then decide based on the context summary whether to react and what that reaction would be.
>
> ### **Lack of objective metric**
> **Q:** There are no objective evaluation metrics for Lyfe Agents other than cost per hour (which, as stated in point 3, is an unfair comparison). The police success rate shown in Figure 3b is absurd. It is based on a self-designed script with predetermined and clear answers, which is not convincing and should not serve as an evaluation metric in a scientific paper.
>
> **A:** As mentioned above, we are evaluating other agents in this environment now. We hope these new experiments show that the scenarios we designed are not trivial for other agents.
>
> We want to clarify what the reviewer means here. The agent backstories are provided in the appendix where only a single agent has the memory available to infer the murderer. The task is then an evaluation of information spreading to the right agent, namely the detective Lizhi. While the backstory is preset, the agents are not scripted. Their action selection is naturally stochastic, due to the stochasticity of GPT. If the reviewer is concerned that the task is trivial, we want to emphasize that this was the objective of our ablation analysis. For additional comparisons, as mentioned earlier, we are investigating the inclusion of the Park et al. memory and retrieval mechanism. More updates on that are to come.

---

> ### Author Response · Authors · 2023-11-11
> **Addressing additional questions**
>
> ### **Additional questions**
> **Q:** What is the problem (motivation) addressed in this paper? llm-based agents with too many queries? What is the factor leading to the "extensive queries" of existing LLM-based agents? Are the queries really extensive? Is the memory mechanism resulting in the "extensive queries"?
>
> **A:** As mentioned above, LLM-based agents often do use many more queries that are necessary. As an example, mentioned above, Park et al’s method for providing a context summary requires three language model calls, whereas our approach of using a rolling update via self-monitor uses much less (we reuse the context and update less frequently since the general situational context doesn’t change that rapidly). We will rewrite our introduction to make this clearer.
>
> **Q:** How do the proposed Lyfe agents solve the problems addressed by the existing LLM-based agents? Which module contributes the most to efficiency?
>
> **A:** We will rewrite the modules section to better explain how the modules support high performance without a high cost.
>
> **Q:** How are the scripts (scenario 1 & 2) driven?
>
> **A:** Agents are provided with an initial goal, a set of long-term memories that provide agent identity, and recent memories that provide the initial context  (see Appendix G for Murder Mystery). There are no scripts, just the initial conditions are set. Then the agents are run, being asked to generate an action at each step, according to the rules and architecture outlined in the paper.
>
> **Q:** What is the LLM used in the experiments?
>
> **A:** LLM is GPT3.5-turbo, but we are also running new experiments using Llama-2-70B, Mistral-7B, and GPT4.
>
> **Q:** Could existing LLM-based agents be real-time agents if the LLM is running locally?
>
> **A:** The real-time nature is largely due to our more efficient parallel processing infrastructure, but we realize that this is not really a point worth highlighting for this academic paper, so we are removing the emphasis on real-time from the paper.

---

> ### Author Response · Authors · 2023-11-22
>
> Dear Reviewer 4MNZ,
>
> We appreciate your insightful comments and suggestions. We would like to present updated results from our new simulations after modifications and look forward to further discussions.
>
> We will continue to work on it as some of these results are preliminary. The items are (1) comparing the performance of our agents with other language models, (2) cost comparison of our agent memory vs. an implementation using Stanford’s memory, and (3) progress on the overall structure of the paper. For (1), we find that GPT3.5 and GPT4 have comparable performance whereas open-source models tend to do worse. For (2), we find that our memory uses about 30X fewer tokens per action selection. For (3), we are in the middle of changes to the structure, having already modified our literature review for better contextualization and the introduction to emphasize that our agents are intended for social interaction. Further details on these items are provided below.
>
> **1. Running simulations for other language models.**
>
> We ran our simulations using GPT4, Mistral-7B, and Llama-70B and present the results of our preliminary exploration. To normalize these comparisons, we modified the simulation length for each one so that they had approximately the same number of action selection decisions per simulation. We preface that these results are preliminary, we mention some issues below. The results of each trial:
>
> | Model   | Mean ± Standard Error |
> |---------|-----------------------|
> | gpt3.5-Turbo   | 0.58 ± 0.08            |
> | gpt4   | 0.44 ± 0.09            |
> | llama2-70B   | 0.13 ± 0.06            |
> | mistral-7B   | 0.23 ± 0.09            |
>
>
> The GPT3.5 runs include the simulations from the original submission along with some additional runs we made during the rebuttal period. We note that there is a caveat for Mistral and Llama, the percentage of incorrectly formatted outputs was higher. This led to requiring longer runs (since effective action selection takes longer to get right) and there may be other confounding factors contributing to the performance.
>
> **2. Comparison of cost with Stanford**
>
> To make a better comparison of our cost with that of the architecture from Park et al., we took their memory and implemented it in our agent for comparison. We kept our self-monitoring summary and action selection modules, but replaced our memory. Our runs indicate that our memory utilizes approximately 30 times fewer tokens than that of the Park et al. memory. Due to differences in latency, we normalize the comparison by looking at the rate of token cost for the memory per action-selection. For our base architecture, the cost was 22.00 tokens per action-selection whereas for Park et al. we found 612.53 tokens per action-selection. We note that the performance using Park et al.’s architecture was also not great, though we anticipate this may be due to our manner of defining memory keywords for which there wasn’t a straightforward adaptation from Stanford to our setting. We plan to further investigate this.
>
> **3. Modifications to the paper**
>
> We have incorporated several changes to the paper, though this is ongoing. The introduction better emphasizes the aim of this work as developing agents for social simulation and the relation between the modules and cost effectiveness, with details in Section 3 --- more on this in the next paragraph. We moved the literature review to follow the introduction, better contextualizing our work amidst them. In particular, we discuss language model powered agents, existing benchmarks and problems therein, long-term memory via vector database, and coherent goal-directed behavior with reflection. More details will be provided on model architecture.
>
> We are still in progress of making these changes, but want to summarize the argument for the cost-effectiveness of each module. For action-selection our argument remains the same, to save calls on choosing higher level actions through the hierarchical design. We plan to incorporate our investigations from item (2) above to demonstrate our consideration for cost in memory. We also emphasize the cost-effectiveness and novelty of the self-monitoring summary as a way to provide context for action-selection. In particular, we point out that this provides context for action-selection with a slow update that allows for reuse across multiple action steps — as the contextual narrative changes slowly relative to action steps.
>
>
> Sincerely
> Submission 8143

---

> > ### Comment · Reviewer_4MNZ · 2023-11-22
> >
> > I would like to express my gratitude for the response from the authors. They have demonstrated a constructive attitude towards the reviews and have made significant improvements in the manuscript.
> >
> > However, my questions on the points 2, 3, 5, 6, and the point 2 of questions  have not been addressed well, therefore, i will retain my score.
> >
> > The response from the authors even raises more problems of this paper.
> >
> > 1. The performance of the proposed Lyfe, powered by ChatGPT 3.5, is significantly better than that of ChatGPT 4. This difference highlights the overfitting issue in the proposed agent. As I mentioned in my initial review, the metric of the police success rate presented in Figure 3b is questionable. The experiments appear to be merely theoretical exercises designed by the authors, lacking convincing real-world applicability.
> >
> > 2. A fair comparison of a general agent should involve scenarios specifically designed for such agents, rather than the text-based exercises proposed by the authors. It is problematic to be both the creator of the questions and the evaluator of the answers. This practice leads to an overfitting issue, resulting in an unfair advantage for Lyfe in these text-based scenarios. The text game is a kind of Murder Mystery Game that played by agents only.
> >
> > 3. I concur with the reviewer Qy47 in that the inclusion of a 3D environment does not add value to this paper. I also question whether the authors actually developed the code for the agent to interact with the 3D scenarios, or if they simply manually simulated these interactions based on outputs from the Large Language Model (LLM).
> >
> > 4. The response from the authors should adhere to the formatting standards of Markdown syntax for clarity and consistency (Table Results).

---

> > > ### Author Response · Authors · 2023-11-22
> > > **Please retract the comment questioning our integrity**
> > >
> > > [This is not a public comment]
> > >
> > > The reviewer asked "I also question whether the authors actually developed the code for the agent to interact with the 3D scenarios, or if they simply manually simulated these interactions based on outputs from the Large Language Model (LLM)."
> > >
> > > We mentioned many times in our paper that our environment supports **real-time** human-agent interactions, and we show the human-agent interaction prominently in Figure 1. To question whether we have developed the code is simply an insult. It takes a tremendous amount of engineering effort to support real-time interactions. This is a scientific paper so we didn't describe the engineering effort necessary, but this is not a reason to essentially accusing us of cheating. We ask the reviewer to retract this specific comment.
> > >
> > > Here's my final take on this:
> > > Look, it's one thing to get your paper scored poorly or rejected, and that's fine, happens all the time. But in my entire career, I have never had a single reviewer suggests that we are making up results (not to mention with no evidence and reasoning at all). This is a serious allegation.

---

> ### Author Response · Authors · 2023-11-22
> **Responding to reviewer**
>
> 1. We are sorry for not providing better formatted results. Fixed. Statistically speaking, GPT4 result is not significantly different from GPT3.5 (p=0.28 without correction). In fact, accounting for multi-comparison, only Llama-2 is significantly worse than GPT3.5 due to the large variance across runs.
>
> Continued. We agree that it's limited that we use a scenario that we designed ourselves. However, this field is very new. As of April 2023, the state-of-the-art in this area (Park et al 2023) reported results from a single 2 game-day run. Here we are already reported results from many more runs with different LLMs usage.
>
> 2. Again, we agree that it's not ideal to be the player and the judge at the same time. But there is **no standardized benchmarks in this field**, and not to mention using our own environment. For example, the reviewer cites AgentSims, which has no benchmark in it.
>
> 3. This question is an insult to our integrity. We agree that the 3D environment is not critical to the scientific message here (a point we agreed already in our response to Qy47), but we spent significant engineering effort to develop the 3D environment so humans can interact with agents directly in this environment. We already showed in Figure 1 that a human player is interacting with these agents, and we had put real-time in our title of the paper. We ask the reviewer to retract this specific comment.
>
> 4. Thank you. Fixed

---

> ### Author Response · Authors · 2023-11-22
> **Reviewer Response (2/2)**
>
> **Q:** However, my questions on the points 2, 3, 5, 6, and the point 2 of questions have not been addressed well, therefore, i will retain my score.
>
> We already replied to these individual points a week ago, but the reviewer didn't give any comment up until we have less than 24 hours to respond. The results posted today agree with our argument from a week ago.
>
> **Point 3** is comparison with existing LLM-based agents. The current state of LLM-based agents is that they are generally not inter-operable, unlike LLMs or computer vision models. For example, the influential Park et al model, which we studied closely, cannot be easily applied to our environment because its agent source code is deeply intertwined with the specific environment setup they have (called maze in their code). Another influential work Voyager for Minecraft relies on code-execution for low level actions. The examples go on.
>
> **Point 5** is suggesting we change our schematic figure to a general LLM-based agent figure. We simply disagree. This is like asking a computer vision paper to change their schematic figure to a general convnet schematic like ResNet, this doesn't help the readers.
>
> **Point 5** is also suggesting that we evaluate our agents in well-known scenarios [3, 4]. We already explained why that's not appropriate. Similarly, agents developed in [3] and [4] cannot be applied to each other's scenario.
>
> **Point 6** is about comparing our self-monitoring mechanism with Park's reflection mechanism. First Park's reflection mechanism is more related to the summarize-and-forget memory mechanism we proposed. Second, we did compare the two mechanisms and showed that Park's mechanism costs 30X more tokens in our environment with no clear benefit.

---

> > ### Comment · Reviewer_4MNZ · 2023-11-22
> >
> > I would like to express my gratitude for the response from the authors. However, the response from the authors still does not resolve the problems.
> >
> > 1. My comment was "I also question whether the authors actually developed the code for the agent to interact with the 3D scenarios", instead of the development of the 3D games. Given that ChatGPT 3.5 is limited to processing text-based information, it raises the question: how can it perceive 3D elements such as positions and directions? Additionally, how does the language model (LLM) control characters in 3D games? Without the capability to understand 3D positions and directions, what role does the interface play for human players? I would appreciate it if the authors consider these aspects carefully before responding.
> >
> > 2. The core issue lies in the authors simultaneously assuming the roles of question creators and evaluators of the answers. This dual role could introduce bias. A more objective approach would involve comparing their agents against benchmarks established by others. Utilizing external benchmarks would likely yield a fairer and more credible evaluation.
> >
> > 3. Why the authors alter the experiment data?
> > The original results for GPT 3.5 and GPT4 are
> > |   |   |   |   |   |   |   |   |   |   |
> > |---|---|---|---|---|---|---|---|---|---|
> > | GPT3.5|  0.852 |	0.667|	0.556|0.630	|0.407|	0.778|	0.593|	0.037|	0.741|
> > |GPT4 | 0.370 |	0.741 |	0.222 |	0.556 |	0.296 | | | | | |
> >
> > The flutuations are very serious, and the mean and standard deviation should be
> > |   |   |
> > |---|---|
> > | GPT3.5| 0.58 $\pm$ 0.23 |
> > |GPT4 | 0.44 $\pm$ 0.19 |
> >
> > However, the authors' provided standard deviation is much smaller, shown as below
> > |   |   |
> > |---|---|
> > | GPT3.5| 0.58 $\pm$ 0.08 |
> > |GPT4 | 0.44 $\pm$ 0.09 |
> >
> > Can the authors explain this ?

---

> ### Author Response · Authors · 2023-11-22
> **Standard error of the mean is not Standard deviation**
>
> 1. Of course we "developed the code for the agent to interact with the 3D scenarios". LLM-based agents can still be used in 3D environments. Similar to our work, Voyager (which the reviewer cited) used the 3D Minecraft environment and use LLM-based agent. We never claimed that the agents have image-based visual inputs.
>
> Again, the 3D environment is not part of our core contributions, and we added this to the main text: "the 3D nature of this environment is not pertinent to the scientific results of this work. The same results can in principle be obtained in 2-D environments, or even pure text-based environments (with proper navigation and group chat dynamics set up)".
>
> We understand that we did not provide enough details to fully understand how the agents work in the environment. Please understand that we are introducing (1) a new multi-agent environment, (2) three different techniques for the agent within a single paper so we simply don't have space to describe all the details in the main text or even the Appendix.
>
> 2. We agree it's not the best to make both the agent and the benchmark. Again, this field is very new, as I'm sure the reviewer realizes, and there are no standard benchmarks for social agents (essentially no benchmark at all). We already explained that. Stanford Generative Agent, AgentSims, Tsinghua S3, etc., none of them have benchmarks. I'm not saying we don't want benchmarks. I'm just saying that standard benchmarks haven't been developed yet.
>
> 3. Please stop casually making serious accusations like "alter the experiment data" (which is a fireable offense).
> We are reporting standard error of the mean (SE), not standard deviation (STD). SE and STD are both informative, but SE is more useful for eye-balling whether two mean values are significantly different (doesn't replace statistical test, which we did). We had already put Standard Error in the title of that column

---

### Official Review · Reviewer_yeom · 2023-10-30

**Soundness:** 3 good
**Presentation:** 3 good
**Contribution:** 2 fair
**Rating:** 5
**Confidence:** 3

**Summary:**

This paper introduces Lyfe Agents, an LLM framework for simulating social behaviour in virtual societies. In terms of the LLM framework, the paper proposes 3 new architecture components: a module for high-level decision-making, a module for helping agents maintain contextual awareness, and an improved memory module. In addition, the paper introduces a new environment, LyfeGame, for studying social interactions amongst LLMs.

**Strengths:**

- I like how the authors have grounded their three new modules in brain-inspired research. All three of the modules appear to be valid and useful in terms of improving performance of the agents, and I appreciate that they are also designed in order to be cost effective.
- I think the scenarios that the authors have developed in order to evaluate their agents are smart and well-designed, and useful for picking out interesting emergent behaviours amongst LLM agents. In particular the murder mystery scenario, and the corresponding ablation that demonstrates the importance of all of the proposed modules.

**Weaknesses:**

I have two main problems with the paper that, in my opinion, need to be addressed.

1) Whilst the intentions and role of all of the modules are well explained, the details of their implementations are not particularly well explained within the main body of the text. For example, take the option-action selection module: what are the implementation details for the LLM call that is used to output an options along with a subgoal? What are the available options, is this defined by the LLM or a discrete list provided beforehand? What are the details of the exit conditions that lead to the option-action selection module being called? Whilst there are some more details included in the appendix, I believe some more important implementation points should be included in the main body of the text.

2) I am not currently convinced by the cost analysis that is provided by the authors. This is particularly important as one of the key-selling points of the framework, according to the authors, is the cost effectiveness of the framework. I am not disputing the project cost of the Stanford GenAgent work. However, I just have a few queries about the cost provided for the Lyfe Agent. Firstly, is the difference in cost primarily coming from the fact that GPT3.5 is used for LyfeAgent vs. GPT-4 for Stanford GenAgent? In my opinion, this isn't necessarily a fair comparison - instead would it not be fairer to evaluate more in terms of tokens inputted / outputted when solving the scenarios? For example, how many tokens need to be inputted / outputted by LyfeAgent vs. Stanford GenAgent in order to arrive at a Murder mystery solution? In my opinion this is a fairer comparison.

**Questions:**

I would be grateful if the authors could address my points brought up in the weaknesses. Primarily:

1) Please provide more implementation details for the modules, especially in the main body of the paper. Currently, for example, I would not personally be able to reproduce this framework based on all the information provided in the paper and appendix.

2) Please discuss my thoughts on the cost-analysis. Is my point fair? If the point relates to GPT3.5 vs. GPT4, please discuss why it is fair to compare the two.

I would be happy to update my score based on the authors responses to these points.

---

> ### Author Response · Authors · 2023-11-11
> **Thank you for the very thoughtful and constructive feedback!**
>
> ### **Adding implementation details in the main text**
> **Q:** Whilst the intentions and role of all of the modules are well explained, the details of their implementations are not particularly well explained within the main body of the text.
>
> **A:** We agree. We are adding substantially more technical details of the key components into the **main text** and Appendix, and will better contextualize these components with the literature.
>
> ### **Better cost analysis**
> **Q:** I am not currently convinced by the cost analysis that is provided by the authors…isn't necessarily a fair comparison…fairer to evaluate more in terms of tokens inputted / outputted when solving the scenarios
>
> **A:** We will add a detailed account of token usage for different components.
>
> Thank you for pointing this out. We agree that comparing across different LLM-based agents can often be apples and oranges, because the environments/tasks are very different. We are planning to look into adapting Stanford GenAgent’s memory and retrieval architecture into our agents as a better point of comparison, and relying on token-based cost for comparison rather than purely USD. We note that the Stanford GenAgent architecture is interwoven with the environment, so this adaptation may face difficulties, but we will regardless add a **new discussion section that explains why it’s difficult to directly compare them fairly**. The low-cost nature of our work remains true, but we will not emphasize it by comparing it with the Stanford paper. (Stanford paper did not use GPT-4 though, they used text-davinci-3, which is still more expensive)

---

> ### Author Response · Authors · 2023-11-22
>
> Dear Reviewer yeom,
>
> We appreciate your insightful comments and suggestions. We would like to present updated results from our new simulations after modifications and look forward to further discussions.
>
>
> **Adding implementation details in the main text**
>
> To enhance the clarity of our manuscript, we are actively working on better contextualizing our key components, incorporating additional technical details and relevant literature.
>
> We have incorporated several changes to the paper, though this is ongoing. The introduction better emphasizes the aim of this work as developing agents for social simulation and the relation between the modules and cost effectiveness, with details in Section 3 --- more on this in the next paragraph. We moved the literature review to follow the introduction, better contextualizing our work amidst them. In particular, we discuss language model powered agents, existing benchmarks and problems therein, long-term memory via vector database, and coherent goal-directed behavior with reflection. More details will be provided on model architecture.
>
> **Better cost analysis**
>
> To make a better comparison of our cost with that of the architecture from Park et al., we took their memory and implemented it in our agent for comparison. We kept our self-monitoring summary and action selection modules, but replaced our memory. Our runs indicate that our memory utilizes approximately 30 times fewer tokens than that of the Park et al. memory. Due to differences in latency, we normalize the comparison by looking at the rate of token cost for the memory per action-selection. For our base architecture, the cost was 22.00 tokens per action-selection whereas for Park et al. we found 612.53 tokens per action-selection. We note that the performance using Park et al.’s architecture was also not great, though we anticipate this may be due to our manner of defining memory keywords for which there wasn’t a straightforward adaptation from Stanford to our setting. We plan to further investigate this.
>
> We are still in progress of making these changes, but want to summarize the argument for the cost-effectiveness of each module. For action-selection our argument remains the same, to save calls on choosing higher level actions through the hierarchical design. We plan to incorporate our investigations from comparison of cost with Stanford above to demonstrate our consideration for cost in memory. We also emphasize the cost-effectiveness and novelty of the self-monitoring summary as a way to provide context for action-selection. In particular, we point out that this provides context for action-selection with a slow update that allows for reuse across multiple action steps — as the contextual narrative changes slowly relative to action steps.
>
>
>
> Sincerely,
>
> Submission8143

---

> > ### Comment · Reviewer_yeom · 2023-11-22
> >
> > I thank the authors for taking the time to answer my questions.
> >
> > Point 1. I am glad to hear that the authors are taking the time to improve the clarity of the manuscript. I would potentially be willing to increase my score if I got the chance to see these changes, however because I believe the changes needed are fairly substantial I am not willing to do so without seeing the edits.
> >
> > Point 2. I agree with the authors that the results for cost analysis look promising, albeit a little too promising. I think the authors raise a valid point when saying that there are concerns with the exact implementation of certain aspects (e.g. memory keywords), which raises the validity of the result. Again, I think the authors are on the right track in terms of what they believe should be added to the paper in order to improve it, however without seeing these changes I can not improve my score.

---

> > > ### Author Response · Authors · 2023-11-22
> > > **Fair point re: clarity**
> > >
> > > I think your comment re: clarify is fair.
> > >
> > > It took us a *lot* of time to incorporate Park's memory structure into our system. Last 10 days we spent most of our time doing this plus rewriting the Intro+Related work plus running experiments on open-sourced LLMs (coding that up was easy, understanding why their results were so poor took more time). So we didn't get to rewrite the Methods section and add more details as we intended too. I understand your reservation and frankly I'd feel the same.
> > >
> > > About the cost analysis though, our result is not really too good to be true. Park et al is a great paper (a lot of respect for them), but it's really not focused on saving cost. The memory system there uses a LLM call to assign importance score to *every* memory item, whereas we use LLM to organize the memory only once in a while, that's why it's 30X cheaper for the memory part.

---

### Official Review · Reviewer_Qy47 · 2023-10-31

**Soundness:** 3 good
**Presentation:** 3 good
**Contribution:** 2 fair
**Rating:** 5
**Confidence:** 4

**Summary:**

The authors introduce generative agents based on ChatGPT, combining a modular internal architecture inspired by neurobiology. As a core concept, they employ a self-monitoring mechanism to align the agent's behavior with their internal goals. It combines a textual representation of recent events and the internal motivation to plan the next steps. In addition, each agent has a dynamic memory that changes based on perceived inputs and events. This module is divided into short and long-term containers. While inputs are first stored in the short-term memory, they can be clustered and summarized into the long-term. The work overserves and evaluates the agent’s behavior in a custom 3D virtual environment in three simulated scenarios: solving an (easy, hard) murder mystery, discussing joining social clubs, and finding a suitable treatment for a medical issue. In addition, the authors perform different ablation scenarios to evaluate the performance of the subparts.

**Strengths:**

•	Agent “Brain”: The authors combine and enhance different modeling techniques that are motivated by the human brain. This approach shows promising results in the simulation of emergent social behavior. I want to highlight the combination of the self-observation and the dynamic memory system. In combination with the retrieval process based on vector similarity, I find this a fair contribution to the conference.
•	Multiple Experiments + Ablation: The work contains three experiments covering different domains of collaboration and conversation. The results show that the agents successfully manage all scenarios. In addition, the ablation setup provides insights into the importance of each submodule in the larger context. Thus, I find the results sound in terms of technical claims and experimentation setup.

**Weaknesses:**

•	Implementation of the virtual world: I see no benefit in highlighting the 3D environment work this work contribution. The interaction is purely text-based, and the agents do not perceive their visuals as image inputs. Thus, the work could be reduced to a text-based role-play scenario.
•	Appendix Structure: I found relevant information about the agent architecture missing in the content but given inside the appendix. Thus, moving relevant information into the body of the paper would emphasize the main contribution.
•	Reliance on GPT3.5: Like previous agent-based work (Park et al.), the observations and consequences are limited to a singular foundation model. As this work employs complex memory and decision mechanisms, the behavior of competitive models like Llama-2 70B or Falcon 180B, when prompted with this approach, may strengthen impact.

**Questions:**

I would like to see examples of the prompt templates used to interact with ChatGPT. Further, I suggest focusing on the agent’s architecture in the main content instead of utilizing the space for a half-page screenshot of the environment. The quality and relevance of the paper would improve with more focus on the contribution regarding the learning representation itself: memory architecture, discerning storage, and retrieval.

**Details Of Ethics Concerns:**

While the names of the individual agents seem to present people from various heritages (Western and Eastern), the provided long-term memories focus on Japanese stereotypes: Dream of moving from the countryside to Tokyo, Loving Mangas, Bonsais, Kendo, or Cherry Blossoms. We see this only as a minor issue, as the authors also focus on an anime-like 3D environment, which seems sound. However, this might restrict the impact of the work.

---

> ### Author Response · Authors · 2023-11-11
> **Thank you for the careful and helpful feedback!**
>
> We are confident that we can address all the concerns. We are running more simulations and rewriting much of the paper, see below.
>
> ### **Implementation of the virtual world**
> **Q:** I see no benefit in highlighting the 3D environment contribution in this work. The interaction is purely text-based, and the agents do not perceive their visuals as image inputs. Thus, the work could be reduced to a text-based role-play scenario
>
> **A:** We agree. The 3D environment makes it more appealing and intuitive for human users, and supports the use of image-based or ray-cast visual inputs, which is motivation for future work. You are right that 3D shouldn’t be seen as the selling point for this work (not our intention either). We are adding this to the main text:
> “The 3D nature of this environment is not pertinent to the scientific results of this work. The same results can in principle be obtained in any environment which supports navigation and group chat dynamics.
>
> ### **More implementation information in main text**
> **Q:** relevant information about the agent architecture (memory architecture, discerning storage, and retrieval, prompt templates) missing
>
> **A:** We are working on adding substantially more technical details of the key components to the main text, and will better contextualize these components with the literature. We will also make space by reducing the size of Figure 1 (environment).
>
> ### **Reliance on GPT3.5**
> **Q:** Like previous agent-based work (Park et al.), the observations and consequences are limited to a singular foundation model…
>
> **A:** We are currently **running more simulations** replacing GPT3.5-turbo with Llama-2-70B, Mistral-7B, and GPT4.
>
> ### **Potential diversity concerns**
> **Q:** …long-term memories focus on Japanese stereotypes...
>
> **A:** We agree with the reviewers. These Japanese-styled background stories are irrelevant to the scientific content of this work, therefore we are **rerunning simulations** after removing these Japanese-specific memories.

---

> ### Author Response · Authors · 2023-11-22
>
> Dear Reviewer Qy47,
>
> We appreciate your insightful comments and suggestions. We would like to present updated results from our new simulations after modifications and look forward to further discussions.
>
> We have modified and removed Japanese-specific memories, and rerun simulations.
>
> We ran our simulations using **GPT3.5, GPT4, Mistral-7B, and Llama-70B** and presented the results of our preliminary exploration. To normalize these comparisons, we modified the simulation length for each one so that they had approximately the same number of action selection decisions per simulation. We preface that these results are preliminary, we mention some issues below. The results of each trial:
>
> | Model   | Mean ± Standard Error |
> |---------|-----------------------|
> | gpt3.5-Turbo   | 0.58 ± 0.08            |
> | gpt4   | 0.44 ± 0.09            |
> | llama2-70B   | 0.13 ± 0.06            |
> | mistral-7B   | 0.23 ± 0.09            |
>
>
> The GPT3.5 runs include the simulations from the original submission along with some additional runs we made during the rebuttal period. We note that there is a caveat for Mistral and Llama, the percentage of incorrectly formatted outputs was higher. This led to requiring longer runs (since effective action selection takes longer to get right) and there may be other confounding factors contributing to the performance.
>
> To enhance the clarity of our manuscript, we are actively working on better contextualizing our key components, incorporating additional technical details and relevant literature.
>
> For a comprehensive overview of the changes made, please refer to our detailed comment at [comment from authors](https://openreview.net/forum?id=VaZa8zj0Yw&noteId=N5LqxZ1RW2).
>
>
>
>
>
>
> Sincerely,
>
> Submission8143

---

### Public Comment · ~Guohao_Li1 · 2023-11-14
**Suggesting related work**

The authors introduce Lyfe Agents, a novel approach to developing generative agents capable of simulating complex social behaviors in virtual societies. These agents, known as Lyfe Agents, are designed to be low-cost, real-time responsive, intelligent, and goal-oriented. Key innovations include an option-action framework, asynchronous self-monitoring for self-consistency, and a Summarize-and-Forget memory mechanism. The agents were evaluated across multiple scenarios in a custom 3D virtual environment, LyfeGame, demonstrating their ability to simulate human-like social reasoning at a computational cost significantly lower than existing alternatives.

Thanks for the great work! It could also be beneficial to discuss prior work on multi-LLM agents for the society study under cooperative settings [1].

[1] Li, Guohao, Hasan Abed Al Kader Hammoud, Hani Itani, Dmitrii Khizbullin, and Bernard Ghanem. "CAMEL: Communicative Agents for" Mind" Exploration of Large Language Model Society." NeurIPS 2023

---

### Author Response · Authors · 2023-11-21
**Updates on additional simulations and paper changes as per reviewers' requests**

We are providing a general response to all reviewers due to shared concerns. There are three items which address feedback from the reviewers. We will continue to work on it as some of these results are preliminary. The items are (1) comparing the performance of our agents with other language models, (2) cost comparison of our agent memory vs. an implementation using Stanford’s memory, and (3) progress on the overall structure of the paper. For (1), we find that GPT3.5 and GPT4 have comparable performance whereas open-source models tend to do worse. For (2), we find that our memory uses about 30X fewer tokens per action selection. For (3), we are in the middle of changes to the structure, having already modified our literature review for better contextualization and the introduction to emphasize that our agents are intended for social interaction. Further details on these items are provided below.

1. Running simulations for other language models.

We ran our simulations using GPT4, Mistral-7B, and Llama2-70B and present the results of our preliminary exploration. To normalize these comparisons, we modified the simulation length for each one so that they had approximately the same number of action selection decisions per simulation. We preface that these results are preliminary, we mention some issues below. The results of each trial:

| Model   | Mean ± Standard Error |
|---------|-----------------------|
| gpt3.5-Turbo   | 0.58 ± 0.08            |
| gpt4   | 0.44 ± 0.09            |
| llama2-70B   | 0.13 ± 0.06            |
| mistral-7B   | 0.23 ± 0.09            |

The GPT3.5 runs include the simulations from the original submission along with some additional runs we made during the rebuttal period. We note that there is a caveat for Mistral and Llama, the percentage of incorrectly formatted outputs was higher. This led to requiring longer runs (since effective action selection takes longer to get right) and there may be other confounding factors contributing to the performance. The difference between GPT3.5 and GPT4 is not significant due to the large variance across runs.

2. Comparison of cost with Stanford

To make a better comparison of our cost with that of the architecture from Park et al., we took their memory and implemented it in our agent for comparison. We kept our self-monitoring summary and action selection modules, but replaced our memory. Our runs indicate that our memory utilizes approximately 30 times fewer tokens than that of the Park et al. memory. Due to differences in latency, we normalize the comparison by looking at the rate of token cost for the memory per action-selection. For our base architecture, the cost was 22.00 tokens per action-selection whereas for Park et al. we found 612.53 tokens per action-selection. We note that the performance using Park et al.’s architecture was also not great, though we anticipate this may be due to our manner of defining memory keywords for which there wasn’t a straightforward adaptation from Stanford to our setting. We plan to further investigate this.

3. Modifications to the paper

We have incorporated several changes to the paper, though this is ongoing. The introduction better emphasizes the aim of this work as developing agents for social simulation and the relation between the modules and cost effectiveness, with details in Section 3 --- more on this in the next paragraph. We moved the literature review to follow the introduction, better contextualizing our work amidst them. In particular, we discuss language model powered agents, existing benchmarks and problems therein, long-term memory via vector database, and coherent goal-directed behavior with reflection. More details will be provided on model architecture.

We are still in progress of making these changes, but want to summarize the argument for the cost-effectiveness of each module. For action-selection our argument remains the same, to save calls on choosing higher level actions through the hierarchical design. We plan to incorporate our investigations from item (2) above to demonstrate our consideration for cost in memory. We also emphasize the cost-effectiveness and novelty of the self-monitoring summary as a way to provide context for action-selection. In particular, we point out that this provides context for action-selection with a slow update that allows for reuse across multiple action steps — as the contextual narrative changes slowly relative to action steps.

---

> ### Comment · Reviewer_tWHo · 2023-11-22
> **Question regarding the new result**
>
> Thank you very much for the additional simulations, and I was wondering how to interpret the results. Are these numbers average success rates? Why are there multiple numbers for each model? If they are success rates for different runs, what is a potential reason for such high variances?

---

> ### Author Response · Authors · 2023-11-22
>
> We replaced the confusing numbers with a clear table of mean success rate with its standard error.
>
> **Q: What is a potential reason for such high variances?**
>
> **A:** That is a great question. Based on our understanding of the simulation dynamic and agents behaviors, there are multiple factors for the stochasticity. For example, different participants for multi-agents group chat will results in different scale of information diffusion. And when crucial information about "bloody knife" is being spread, other rumors or agents' own opinions are being spread as well, resulting in the complexity and difficulty of distiguishing correct information&fact from others' assumptions. Notably, despite the varied success rates of agents replying with 'the biggest suspect is Francesco,' each agent's response consistently aligns with their individual memory records, demonstrating self-coherence in their interactions.
>
>
> Best,
>
>
> Submission 8143

---

> > ### Comment · Reviewer_tWHo · 2023-11-22
> > **Thank you**
> >
> > Thank you for your additional explanation!
> >
> > Personally, I believe the fact that the effectiveness of the proposed method can only be verified with the OpenAI API, a service whose technical details are not completely disclosed, raises issues regarding reproducibility and transparency. I do not intend to downplay the engineering effort involved in this research, and indeed, I see it as one of the strengths of this study, as I have commented since the first review. However, I personally support research with high reproducibility and transparency, so please understand that it is difficult for me to strongly recommend this study. The fact that the performance of the open-source model has been verified is an important step, and I look forward to this study having a greater impact in the future.

---

> > > ### Author Response · Authors · 2023-11-22
> > > **Thank you**
> > >
> > > We were also surprised to find that Llama2-70B had the worst performance. Our agent relies on the LLM's capability of providing correctly formatted json output, which Llama2 appears to struggle with. It is possible that specific prompt engineering is needed for each LLM but this is out of our scope.
> > >
> > > What you point out about LLM-based agents only working with closed source models is simply the status of our field, and it is out of our (the authors') control. For example, the influential Voyager paper reports that their method only works with GPT-4 ("VOYAGER requires the quantum leap in code generation quality from GPT-4").
> > >
> > > It's unfortunate that open-sourced models do not compete as well, in this and many other cases, with closed-source models, but there is very little we as authors can do about that.

---

> > > > ### Comment · Reviewer_tWHo · 2023-11-22
> > > > **I understood the point.**
> > > >
> > > > I understood that it’s not always easy to make things fully transparent and reproducible in LLM studies. But as clearly stated in the [Author Guide](https://iclr.cc/Conferences/2024/AuthorGuide), reproducibly is one of the critical criteria for ICLR, which I would like to follow. I will not take the lack of reproducibility and transparency the ground for immediate rejection, but they make it hard for me to champion the paper. I hope that we agree to disagree on this point. Let’s wait for AC’s opinion.

---

> > > > > ### Author Response · Authors · 2023-11-22
> > > > > **all good**
> > > > >
> > > > > Dear Reviewer tWHo, you look like a reasonable person and this is not a public comment, so I'll be straightforward.
> > > > >
> > > > > I agree that reproducibility is important.
> > > > >
> > > > > Let's say today we open-source our code, anyone with an openai API key can reproduce our results tomorrow, and it will cost them like 5 USD of compute. If that's not reproducible, I don't know what is. Just because no one can download the model weights of GPT doesn't mean results based on GPT are not reproducible.
> > > > >
> > > > > p.s. If we reject all papers where the model using GPT or Claude or other closed-source LLMs perform better than open-sourced LLMs, then that's only detrimental to academia.

---

> ### Comment · Reviewer_tWHo · 2023-11-22
> **Reproducibility**
>
> I understand the argument presented, though I am not fully convinced honestly.
>
> This research can be reproducible in the short term, but what about a year or five years from now? For instance, if a new study on an LLM Agent were to be submitted to ICLR 2030, how can we guarantee that OpenAI will be providing the same service as it does now? I believe that the reproducibility of research should consider such long-term perspective.
>
> With that said, please take my argument as the viewpoint of someone who is *not* particularly knowledgeable in the LLM field (I had some work on human interaction understanding and that’s probably why this paper was assigned) and whose opinion might be somewhat extreme. Let’s wait for AC to make an appropriate judgment.

---

> > ### Author Response · Authors · 2023-11-22
> > **I hear your point**
> >
> > I understand your concerns about the challenge of reproducing results based on closed-source commercial models. Just this weekend we were worried that we may not be able to use GPT3.5 next week, so the concern is real for sure.
> >
> > Not trying to sway you at this point.
> >
> > It's just going to be difficult for us folks in academia (I hope that's not considered de-anonymizing, and I imagine you are in academia too) if we limit ourselves to the weaker open-sourced LLMs. We already have way less compute than industry so we can't train large models, if we can't even *use* these large models, then we'll be even farther behind.

---

> > > ### Comment · Reviewer_tWHo · 2023-11-22
> > > **need moderation from AC**
> > >
> > > Yes, I understand how challenging it is to achieve the level of LLM achived by the OpenAI API, and how this complicates the reproduciblity issue. Thank you for the extensive discussion. As for me, I tried to sincerely communicates my thoughts, and believe that they have been acknowledged.
> > >
> > > Now I'm thinking that the actual issue about this submission is the lack of moderation from AC, say, in this or other threads. I would recommend consulting directly with the AC from here on.

---

### Meta-Review · Area_Chair_o2r4 · 2023-12-05

**Metareview:**

Reviewers were mainly concerned that this paper was too incremental in light of Park et al. (2023).

**Justification For Why Not Higher Score:**

See discussion point above

**Justification For Why Not Lower Score:**

N/A

---

### Decision · Program_Chairs · 2024-01-16

Reject